# OPTIMISTIC EXPLORATION WITH BACKWARD BOOTSTRAPPED BONUS FOR DEEP REINFORCEMENT LEARNING

## ABSTRACT

Optimism in the face of uncertainty is a principled approach for provably efficient exploration for reinforcement learning in tabular and linear settings. However, such an approach is challenging in developing practical exploration algorithms for Deep Reinforcement Learning (DRL). To address this problem, we propose an Optimistic Exploration algorithm with Backward Bootstrapped Bonus (OEB3) for DRL by following these two principles. OEB3 is built on bootstrapped deep $Q$-learning, a non-parametric posterior sampling method for temporally-extended exploration. Based on such a temporally-extended exploration, we construct an UCB-bonus indicating the uncertainty of $Q$-functions. The UCB-bonus is further utilized to estimate an optimistic $Q$-value, which encourages the agent to explore the scarcely visited states and actions to reduce uncertainty. In the estimation of $Q$-function, we adopt an episodic backward update strategy to propagate the future uncertainty to the estimated $Q$-function consistently. Extensive evaluations show that OEB3 outperforms several state-of-the-art exploration approaches in MNIST maze and 49 Atari games.

## 1 INTRODUCTION

In Reinforcement learning (RL) (Sutton & Barto, 2018) formalized by the Markov decision process (MDP), an agent aims to maximize the long-term reward by interacting with the unknown environment. The agent takes actions according to the knowledge of experiences, which leads to the fundamental problem of the exploration-exploitation dilemma. An agent may choose the best decision given current information or acquire more information by exploring the poorly understood states and actions. Exploring the environment may sacrifice immediate rewards but potentially improves future performance. The exploration strategy is crucial for the RL agent to find the optimal policy.

The theoretical RL offers various provably efficient exploration methods in tabular and linear MDPs with the basic value iteration algorithm: least-squares value iteration (LSVI). *Optimism in the face of uncertainty* (Auer & Ortner, 2007; Jin et al., 2018) is a principled approach. In tabular cases, the optimism-based methods incorporate upper confidence bound (UCB) into the value functions as bonuses and attain the optimal worst-case regret (Azar et al., 2017; Jaksch et al., 2010; Dann & Brunskill, 2015). Randomized value function based on posterior sampling chooses actions according to the randomly sampled statistically plausible value functions and is known to achieve near-optimal worst-case and Bayesian regrets (Osband & Van Roy, 2017; Russo, 2019). Recently, the theoretical analyses in tabular cases are extended to linear MDP where the transition and the reward function are assumed to be linear. In linear cases, optimistic LSVI (Jin et al., 2020) attains a near-optimal worst-case regret by using a designed bonus, which is provably efficient. Randomized LSVI (Zanette et al., 2020) also attains a near-optimal worst-case regret.

Although the analyses in tabular and linear cases provide attractive approaches for efficient exploration, these principles are still challenging in developing a practical exploration algorithm for Deep Reinforcement Learning (DRL) (Mnih et al., 2015), which achieves human-level performance in large-scale tasks such as Atari games and robotic tasks. For example, in linear cases, the bonus in optimistic LSVI (Jin et al., 2020) and nontrivial noise in randomized LSVI (Zanette et al., 2020)

are tailored to linear models (Abbasi-Yadkori et al., 2011), and are incompatible with practically powerful function approximations such as neural networks.

To address this problem, we propose the Optimistic Exploration algorithm with Backward Bootstrapped Bonus (OEB3) for DRL. OEB3 is an instantiation of optimistic LSVI (Jin et al., 2020) in DRL by using a general-purpose UCB-bonus to provide an optimistic $Q$-value and a randomized value function to perform temporally-extended exploration. We propose an UCB-bonus that represents the disagreement of bootstrapped $Q$-functions (Osband et al., 2016) to measure the epistemic uncertainty of the unknown optimal value function. $Q$-value added by UCB-bonus becomes an optimistic $Q^+$ function that is higher than $Q$ for scarcely visited state-action pairs and remains close to $Q$ for frequently visited ones. The optimistic $Q^+$ function encourages the agent to explore the states and actions with high UCB-bonuses, signifying scarcely visited areas or meaningful events in completing a task. We propose an extension of episodic backward update (Lee et al., 2019) to propagate the future uncertainty to the estimated action-value function consistently within an episode. The backward update also enables OEB3 to perform highly sample-efficient training.

Comparing to existing count-based and curiosity-driven exploration methods (Taiga et al., 2020), OEB3 has several benefits. (1) We utilize intrinsic rewards to produce optimistic value function and also take advantage of bootstrapped $Q$-learning to perform temporally-consistent exploration while existing methods do not combine these two principles. (2) The UCB-bonus measures the disagreement of $Q$-values, which considers the long-term uncertainty in an episode rather than the single-step uncertainty used in most bonus-based methods (Pathak et al., 2019; Burda et al., 2019b). Meanwhile, the UCB-bonus is computed without introducing additional modules compared to bootstrapped DQN. (3) We provide a theoretical analysis showing that OEB3 has theoretical consistency with optimistic LSVI in linear cases. (4) Extensive evaluations show that OEB3 outperforms several state-of-the-art exploration approaches in MNIST maze and 49 Atari games.

## 2 BACKGROUND

In this section, we introduce bootstrapped DQN (Osband et al., 2016), which we utilize in OEB3 for temporarily-extended exploration. We further introduce optimistic LSVI (Jin et al., 2020), which we instantiate via DRL and propose OEB3.

### 2.1 BOOTSTRAPPED DQN

We consider an episodic MDP represented as $(\mathcal{S}, \mathcal{A}, T, \mathbb{P}, r)$, where $T \in \mathbb{Z}_+$ is the episode length, $\mathcal{S}$ is the state space, $\mathcal{A}$ is the action space, $r$ is the reward function, and $\mathbb{P}$ is the unknown dynamics. In each timestep, the agent obtains the current state $s_t$, takes an action $a_t$, interacts with the environment, receives a reward $r_t$, and updates to the next state $s_{t+1}$. The action-value function $Q^\pi(s_t, a_t) := \mathbb{E}_\pi \left[ \sum_{i=t}^{T-1} \gamma^{i-t} r_i \right]$ represents the expected cumulative reward starting from state $s_t$, taking action $a_t$, and thereafter following policy $\pi(a_t|s_t)$ until the end of the episode. $\gamma \in [0, 1)$ is the discount factor. The optimal value function $Q^* = \max_\pi Q^\pi$, and the optimal action $a^* = \arg\max_{a \in \mathcal{A}} Q^*(s, a)$.

Deep $Q$-Network (DQN) uses a deep neural network with parameters $\theta$ to approximate the $Q$-function. The loss function takes the form of $L(\theta) = \mathbb{E}[(y_t - Q(s_t, a_t; \theta))^2 | (s_t, a_t, r_t, s_{t+1}) \sim \mathcal{D}]$, where $y_t = r_t + \gamma \max_{a'} Q(s_{t+1}, a'; \theta^-)$ is the target value, and $\theta^-$ is the parameter of the target network. The agent accumulates experiences $(s_t, a_t, r_t, s_{t+1})$ in a replay buffer $\mathcal{D}$ and samples mini-batches in training.

Bootstrapped DQN (Osband et al., 2016; 2018) is a non-parametric posterior sampling method, which maintains $K$ estimations of $Q$-values to represent the posterior distribution of the randomized value function. Bootstrapped DQN uses a multi-head network that contains a shared convolution network and $K$ heads. Each head defines a $Q^k$-function. Bootstrapped DQN diversifies different $Q^k$ by using different random initialization and individual target networks. The loss for training $Q^k$ is

$$L(\theta^k) = \mathbb{E}\left[ \left( r_t + \gamma \max_{a'} Q^k(s_{t+1}, a'; \theta^{k-}) - Q^k(s_t, a_t; \theta^k) \right)^2 \Big| (s_t, a_t, r_t, s_{t+1}) \sim \mathcal{D} \right]. \quad (1)$$

The $k$-th head $Q^k(s, a; \theta^k)$ is trained with its own target network $Q^k(s, a; \theta^{k-})$. If $k$-th head is sampled at the start of an episode when interacting with the environment, the agent will follow $Q^k$ to choose actions in the whole episode, which provides temporally-consistent exploration for DRL.

---

**Algorithm 1** Optimistic LSVI in linear MDP

---
1: **Initialize:** $\Lambda_t \leftarrow \lambda \cdot \mathbf{I}$ and $w_h \leftarrow 0$
2: **for** episode $m = 0$ **to** $M - 1$ **do**
3:     Receive the initial state $s_0$
4:     **for** step $t = 0$ **to** $T - 1$ **do**
5:         Take action $a_t = \arg\max_{a \in \mathcal{A}} Q_t(s_t, a)$ and observe $s_{t+1}$.
6:     **end for**
7:     **for** step $t = T - 1$ **to** $0$ **do**
8:         $\Lambda_t \leftarrow \sum_{\tau=0}^{m} \phi(x_t^\tau, a_t^\tau) \phi(x_t^\tau, a_t^\tau)^\top + \lambda \cdot \mathbf{I}$
9:         $w_t \leftarrow \Lambda_t^{-1} \sum_{\tau=0}^{m} \phi(x_t^\tau, a_t^\tau)[r_t(x_t^\tau, a_t^\tau) + \max_a Q_{t+1}(x_{t+1}^\tau, a)]$
10:        $Q_t(\cdot, \cdot) = \min\{w_t^\top \phi(\cdot, \cdot) + \alpha[\phi(\cdot, \cdot)^\top \Lambda_t^{-1} \phi(\cdot, \cdot)]^{1/2}, T\}$
11:     **end for**
12: **end for**

---

## 2.2 OPTIMISTIC LSVI

Optimistic LSVI (Jin et al., 2020) uses an optimistic $Q$-value with LSVI in linear MDP. We denote the feature map of the state-action pair as $\phi : \mathcal{S} \times \mathcal{A} \to \mathbb{R}^d$. The transition kernel and reward function are assumed to be linear in $\phi$. Optimistic LSVI shown in Algorithm 1 consists of two parts. In the first part (line 3-6), the agent executes the policy according to $Q_t$ for an episode. In the second part (line 7-11), the parameter $w_t$ of $Q$-function is updated in closed-form by following the regularized least-squares problem

$$w_t \leftarrow \arg\min_{w \in \mathbb{R}^d} \sum_{\tau=0}^{m} \left[ r_t(s_t^\tau, a_t^\tau) + \max_{a \in \mathcal{A}} Q_{t+1}(s_{t+1}^\tau, a) - w^\top \phi(s_t^\tau, a_t^\tau) \right]^2 + \lambda \|w\|^2, \quad (2)$$

where $m$ is the number of episodes, and $\tau$ is the episodic index. The least-squares problem has the explicit solution $w_t = \Lambda_t^{-1} \sum_{\tau=0}^{m} \phi(x_t^\tau, a_t^\tau)\left[ r_t(x_t^\tau, a_t^\tau) + \max_a Q_{t+1}(x_{t+1}^\tau, a) \right]$ (line 9), where $\Lambda_t$ is the Gram matrix. Then the value function is estimated by $Q_t(s, a) \approx w_t^\top \phi(s, a)$.

Optimistic LSVI uses a bonus $\alpha[\phi(s, a)^\top \Lambda_t^{-1} \phi(s, a)]^{1/2}$ (line 10) to measure the uncertainty of state-action pairs. We can intuitively consider $u := (\phi^\top \Lambda_t^{-1} \phi)^{-1}$ as a pseudo count of state-action pair by projecting the total features that have observed ($\Lambda_t$) to the direction of the curresponding feature $\phi$. Thus, the bonus $\alpha/\sqrt{u}$ represents the uncertainty along the direction of $\phi$. By adding the bonus to $Q$-value, we obtain an optimistic value function $Q^+$, which serves as an upper bound of $Q$ to encourage exploration. The bonus in each step is propagated from the end of the episode by the backward update of $Q$-value (line 7-11), which follows the principle of dynamic programming. Theoretical analysis (Jin et al., 2020) shows that optimistic LSVI achieves a near-optimal worst-case regret of $\tilde{\mathcal{O}}(\sqrt{d^3 T^3 L^3})$ with proper selections of $\alpha$ and $\lambda$, where $L$ is the total number of steps.

## 3 PROPOSED METHOD

Optimistic LSVI (Jin et al., 2020) provides an atractive approach for efficient exploration. Neverthe-less, developing a practical exploration algorithm for DRL is challenging, since (i) the UCB-bonus utilized by Optimistic LSVI is tailored for linear MDPs, and (ii) optimistic LSVI utilizes backward update of $Q$-functions (line 7-11 in Alg. 1) to aggregate uncertainty. To this end, we propose the following approaches, which are the building blocks of OEB3.

- We propose a general-purpose UCB-bonus for optimistic exploration. More specifically, we utilize bootstrapped DQN to construct a general-purpose UCB-bonus, which is theoretically consistent with optimistic LSVI for linear MDPs. We refer to Section 3.1 for the details.

- We propose a sample-efficient learning algorithm to integrate bootstrapped DQN and UCB-bonus into the backward update, which faithfully follows the principle of dynamic programming. More specifically, we extend Episodic Backward Update (EBU) (Lee et al., 2019) from ordinary $Q$-learning to bootstrapped $Q$-learning, which we abbreviate by BEBU (Bootstrapped EBU). BEBU allows sample-efficient learning and fast propagation of the future uncertainty to the estimated $Q$-value consistently. We further propose OEB3 by combining BEBU and the UCB-bonus obtained via bootstrapped $Q$-learning. We refer to Section 3.2 for the details.

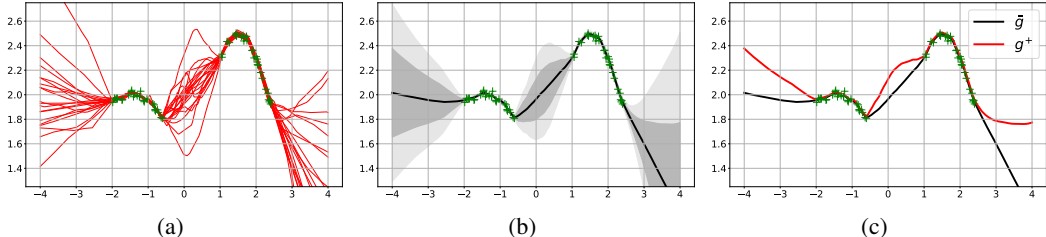

Figure 1: Illustration of UCB-bonus by using bootstrapped estimates in the regression task. The green dots represent 60 data points. (a) Regression curves of 20 neural networks. (b) Mean estimate (black) and uncertainty measurement (shadow). (c) The optimistic value (red) and mean value (black).

### 3.1 UCB-BONUS IN OEB3

Optimistic exploration uses an optimistic action-value function $Q^+$ to incentive exploration by adding a bonus term to the ordinary $Q$-value. Thus $Q^+$ serves an upper bound of ordinary $Q$. The bonus term represents the epistemic uncertainty that results from lacking experiences of the corresponding states and actions. In this paper, we use an UCB-bonus $\mathcal{B}(s_t, a_t)$ by measuring the disagreement of the bootstrapped $Q$-values $\{Q^k(s_t, a_t)\}_{k=1}^K$ of the state-action pair $(s_t, a_t)$ in bootstrapped DQN, which takes the following form,

$$\mathcal{B}(s_t, a_t) := \tilde{\sigma}\left(Q^k(s_t, a_t)\right) = \sqrt{\frac{1}{K}\sum_{k=1}^K \left(Q^k(s_t, a_t) - \bar{Q}(s_t, a_t)\right)^2}, \qquad (3)$$

where $\bar{Q}(s_t, a_t)$ is the mean of bootstrapped $Q$-values. The similar measurement was first used in Chen et al. (2017). We further establish connection between the UCB-bonus defined in Eq. (3) and the bonus in optimistic-LSVI.

**Theorem 1** (Informal Version of Theorem 2). *In linear function approximation, the UCB-bonus $\mathcal{B}(s_t, a_t)$ in OEB3 is equivalent to the bonus-term $[\phi_t^\top \Lambda_t^{-1} \phi_t]^{1/2}$ in optimistic-LSVI, where $\Lambda_t \leftarrow \sum_{\tau=0}^m \phi(x_t^\tau, a_t^\tau)\phi(x_t^\tau, a_t^\tau)^\top + \lambda \cdot \mathbf{I}$, and $m$ is the current episode.*

We refer to Appendix A for the proof and a detailed discussion. Implementing the UCB-bonus $\mathcal{B}(s_t, a_t)$ defined in Eq. (3) in DRL is desirable for exploration for the following reasons.

- Bootstrapped DQN is a non-parametric posterior sampling method, which can be implemented via deep neural networks (Osband et al., 2019).

- The UCB-bonus $\mathcal{B}(s_t, a_t)$ defined in Eq. (3) quantifies the epistemic uncertainty of a specific state-action pair adequately. More specifically, due to the non-convexity nature of optimizing neural network and independent initialization, if $(s_t, a_t)$ is scarcely visited, the UCB-bonus $\mathcal{B}(s_t, a_t)$ obtained via bootstrapped DQN is high. Moreover, the UCB-bonus converges to zero asymptotically as the sample size increases to infinity.

- The UCB-bonus is computed in a batch when performing experience replay, which is more efficient than other optimistic methods that change the action-selection scheme in each timestep (Chen et al., 2017; Nikolov et al., 2019).

The optimistic $Q^+$ is obtained by summing up the UCB-bonus and the estimated $Q$-function, which takes the following form,

$$Q^+(s_t, a_t) := Q(s_t, a_t) + \alpha\mathcal{B}(s_t, a_t). \qquad (4)$$

We use a simple regression task with neural networks to illustrate the proposed UCB-bonus, as shown in Figure 1. We use 20 neural networks with the same network architecture to solve the same regression problem. Each network contains three residual blocks with two fully-connected layers each. According to (Osband et al., 2016), the difference between the outcome of fitting the neural networks is a result of different initialization. For a single input $x$, the networks yield different estimations $\{g_i(x)\}_{i=1}^{20}$ as shown in Figure 1. It follows from Figure 1(a) that the estimations $\{g_i(x)\}_{i=1}^{20}$ behave similar in the region with rich observations, resulting in small disagreement of the estimations, but vary in the region with scarce observations, resulting in large disagreement of the estimations.

In Figure 1(b), we illustrate the confidence bound of the regression results $\bar{g}(x) \pm \tilde{\sigma}(g_i(x))$ and $\bar{g}(x) \pm 2\tilde{\sigma}(g_i(x))$, where $\bar{g}(x)$ and $\tilde{\sigma}(g_i(x))$ are the mean and standard deviation of the estimations. The standard deviation $\tilde{\sigma}(g_i(x))$ captures the epistemic uncertainty of regression results. Figure 1(c) shows the optimistic estimation as $g^+(x) = \bar{g}(x) + \tilde{\sigma}(g_i(x))$ by adding the uncertainty measured by standard deviation. It can be seen that the optimistic estimation $g^+$ is close to $\bar{g}$ in the region with rich observations, and higher than $\bar{g}$ in the region with scarce observations.

In DRL, the bootstrapped $Q$-functions $\{Q^k(s_t, a_t)\}_{k=1}^K$ obtained by fitting the target $Q$-function perform similarly as $\{g_i(x)\}_{i=1}^{20}$ in the regression task. A higher UCB-bonus $\mathcal{B}(s_t, a_t) := \tilde{\sigma}(Q^k(s_t, a_t))$ indicates a higher epistemic uncertainty of the action-value function with $(s_t, a_t)$. The estimated $Q$-function augmented with the UCB-bonus yields $Q^+$, which produces optimistic estimation for novel state-action pairs and behaves similar to the target $Q$-function in areas that are well explored by the agent. Hence, the optimistic extimation $Q^+$ incentives the agent to explore the potentially informative states and actions efficiently.

## 3.2 Uncertainty Backward in OEB3

There are two major reasons for updating the action-value function through Bootstrapped Episodic Backward Update (BEBU) in OEB3.

- The backward propagation utilizes a complete trajectory from the replay buffer. Such an approach allows OEB3 to infer the long-term effect of decision making from the replay buffer. In contrast, DQN and Bootstrapped DQN utilize the replay buffer by random sampling one-step transitions, which loses the information containing such a long-term effect (Lee et al., 2019).

- The backward propagation is required to propagate the future uncertainty to the estimated action-value function consistently via UCB-bonus within an episode. For instance, let $t_2 > t_1$ be the indices of two steps in an episode. If the update of $Q_{t_2}$ occurs after the update of $Q_{t_1}$ within an episode, then it can occur that the uncertainty propagated into $Q_{t_1}$ is not consistent with the uncertainty that $Q_{t_2}$ contains.

To integrate the UCB-bonus into bootstrapped $Q$-learning, we propose a novel $Q$-target in update by adding the bonus term in both the immediate reward and the next-$Q$ value. The proposed $Q$-target needs to be suitable for BEBU in training. Formally, the $Q$-target for updating $Q^k$ is defined as

$$y_t^k := \left[ r(s_t, a_t) + \alpha_1 \mathcal{B}(s_t, a_t; \theta) \right] + \gamma \left[ Q^k(s_{t+1}, a'; \theta^{k-}) + \alpha_2 \mathbb{1}_{a' \neq a_{t+1}} \tilde{\mathcal{B}}^k(s_{t+1}, a'; \theta^-) \right], \quad (5)$$

where $a' = \arg\max_a Q^k(s_{t+1}, a; \theta^{k-})$. The choice of $a'$ is determined by the target $Q$-value without considering the bonus. The immediate reward is added by $\mathcal{B}(s_t, a_t; \theta)$ with a factor $\alpha_1$, where the bonus $\mathcal{B}$ is computed by bootstrapped $Q$-network with parameter $\theta$. The next-$Q$ value is added by $\mathbb{1}_{a' \neq a_{t+1}} \tilde{\mathcal{B}}^k(s_{t+1}, a'; \theta^-)$ with factor $\alpha_2$, where the bonus $\tilde{\mathcal{B}}^k$ is computed by the target network with parameter $\theta^-$. We assign different bonus $\tilde{\mathcal{B}}^k$ of next-$Q$ value to different heads, since the choice of action $a'$ are different among the heads. Meanwhile, we assign the same bonus $\mathcal{B}$ of immediate reward for all the heads. We introduce the indicator function $\mathbb{1}_{a' \neq a_{t+1}}$ to suit the backward update of $Q$-values. More specifically, in the $t$-th step, the action-value function $Q^k$ is updated optimistically at the state-action pair $(s_{t+1}, a_{t+1})$ due to the backward update. Thus, we ignore the bonus of next-$Q$ value in the update of $Q^k$ when $a'$ is equal to $a_{t+1}$.

We use an example to explain the process of backward update. We store and sample the experiences in episodes, and perform update in episodes rather than uniformly sampled transitions. We consider an episode containing three time steps, $(s_0, a_0) \rightarrow (s_1, a_1) \rightarrow (s_2, a_2)$. We thus update the $Q$-value in the head $k$ in the backward manner, namely $Q(s_2, a_2) \rightarrow Q(s_1, a_1) \rightarrow Q(s_0, a_0)$ from the end of the episode. We describe the process as follows.

1. First, we update $Q(s_2, a_2) \leftarrow r(s_2, a_2) + \alpha_1 \mathcal{B}(s_2, a_2)$. Note that in the last time step, we do not need to consider the next-$Q$ value.

2. Then, $Q(s_1, a_1) \leftarrow [r(s_1, a_1) + \alpha_1 \mathcal{B}(s_1, a_1)] + [Q(s_2, a') + \alpha_2 \mathbb{1}_{a' \neq a_2} \tilde{\mathcal{B}}(s_2, a')]$ by following Eq. (5), where $a' = \arg\max_a Q(s_2, a)$. Since $Q(s_2, a_2)$ is updated optimistically in step 1, we ignore the bonus-term $\tilde{\mathcal{B}}$ in next-$Q$ value when $a' = a_2$. The UCB-bonus is augmented in update by adding $\mathcal{B}$ and $\tilde{\mathcal{B}}$ to the immediate reward and next-$Q$ value, respectively.

3. The update of $Q(s_0, a_0)$ follows the same principle. The optimistic $Q$-value is $Q(s_0, a_0) \leftarrow [r(s_0, a_0) + \alpha_1 \mathcal{B}(s_0, a_0)] + [Q(s_1, a') + \alpha_2 \mathbb{1}_{a' \neq a_1} \tilde{\mathcal{B}}(s_1, a')]$, where $a' = \arg\max_a Q(s_1, a)$.

In practice, the episodic update typically leads to instability in DRL due to the strong correlation in consecutive transitions. Hence, we propose a diffusion factor $\beta \in [0, 1]$ in BEBU to prevent instability as in Lee et al. (2019). The $Q$-value is therefore computed as the weighted sum of the current value and the back-propagated estimation scaled with factor $\beta$. We consider an episodic experience that contains $T$ transitions, denoted by $E = \{\mathbf{S}, \mathbf{A}, \mathbf{R}, \mathbf{S'}\}$, where $\mathbf{S} = \{s_0, \ldots, s_{T-1}\}$, $\mathbf{A} = \{a_0, \ldots, a_{T-1}\}$, $\mathbf{R} = \{r_0, \ldots, r_{T-1}\}$ and $\mathbf{S'} = \{s_1, \ldots, s_T\}$. We initialize a $Q$-table $\tilde{\mathbf{Q}} \in \mathbb{R}^{K \times |\mathcal{A}| \times T}$ by $Q(\cdot; \theta^-)$ to store the next $Q$-values of all the next states $\mathbf{S'}$ and valid actions for $K$ heads. We initialize $\mathbf{y} \in \mathbb{R}^{K \times T}$ to store the $Q$-target for $K$ heads and $T$ steps. We use bootstrapped $Q$-network with parameters $\theta$ to compute the bonus $\mathbf{B} = [\mathcal{B}(s_0, a_0), \ldots, \mathcal{B}(s_{T-1}, a_{T-1})]$ for immediate reward, and use the target network with parameters $\theta^-$ to compute bonus $\tilde{\mathbf{B}}^k = [\tilde{\mathcal{B}}^k(s_1, a'_1), \ldots, \tilde{\mathcal{B}}^k(s_T, a'_T)]$ for next-$Q$ value in each head, where $a'_t = \arg\max_a Q^k(s_t, a; \theta^{k-})$. The bonus vector $\mathbf{B} \in \mathbb{R}^T$ is the same for all $Q$-heads, while $\tilde{\mathbf{B}} \in \mathbb{R}^{K \times T}$ contains different values for different heads because the choices of $a'_t$ are different.

In the training of head $k$, we initialize the $Q$-target in the last step by $\mathbf{y}[k, T-1] = \mathbf{R}_{T-1} + \alpha_1 \mathbf{B}_{T-1}$. We then perform a recursive backward update to get all $Q$-target values. The elements of $\tilde{\mathbf{Q}}[k, a_{t+1}, t]$ for step $t$ in head $k$ is updated by using its corresponding $Q$-target $\mathbf{y}[k, t+1]$ with the diffusion factor as follows,

$$\tilde{\mathbf{Q}}[k, a_{t+1}, t] \leftarrow \beta \mathbf{y}[k, t+1] + (1-\beta) \tilde{\mathbf{Q}}[k, a_{t+1}, t]. \tag{6}$$

Then, we update $\mathbf{y}[k, t]$ in the previous time step based on the newly updated $t$-th column of $\tilde{\mathbf{Q}}[k]$ as follows,

$$\mathbf{y}[k, t] \leftarrow (\mathbf{R}_t + \alpha_1 \mathbf{B}_t) + \gamma(\tilde{\mathbf{Q}}[k, a', t] + \alpha_2 \mathbb{1}_{a' \neq a_{t+1}} \tilde{\mathbf{B}}[k, t]), \tag{7}$$

where $a' = \arg\max_a \tilde{\mathbf{Q}}[k, a, t]$. In practice, we construct a matrix $\tilde{\mathbf{A}} = \arg\max_a \tilde{\mathbf{Q}}[\cdot, a, \cdot] \in \mathbb{R}^{K \times T}$ to gather all the actions $a'$ that correspond to the next-$Q$. We then construct a mask matrix $\mathbf{M} \in \mathbb{R}^{K \times T}$ to store the information whether $\tilde{\mathbf{A}}$ is identical to the executed action in the corresponding timestep or not. The bonus of next-$Q$ is the element-wise product of $\mathbf{M}$ and $\tilde{\mathbf{B}}$ with factor $\alpha_2$. After the backward update, we compute the $Q$-value of $(\mathbf{S}, \mathbf{A})$ as $\mathbf{Q} = Q(\mathbf{S}, \mathbf{A}; \theta) \in \mathbb{R}^{K \times T}$. The loss function takes the form of $L(\theta) = \mathbb{E}[(\mathbf{y} - \mathbf{Q})^2 | (s_t, a_t, r_t, s_{t+1}) \in E, E \sim \mathcal{D}]$, where the episodic experience $E$ is sampled from replay buffer to perform gradient descent. The gradient of all heads can be computed simultaneously via BEBU. We refer the full algorithm of OEB3 to Appendix B.

The theory in optimistic-LSVI requires a strong linear assumption in the transition dynamics. To make it works empirically, we make several adjustments in the implementation details. First, in each training step of optimistic-LSVI, all historical samples are utilized to update the weight of $Q$-function and calculate the confidence bonus. While in OEB3, we use samples from a batch of episodic trajectories from the replay buffer in each training step. Such a difference in implementation is imposed to achieve computational efficiency. Second, in OEB3, the target-network has a relatively low update frequency, whereas, in Optimistic LSVI, the target Q function is updated in each iteration. Such implementation techniques are commonly used in most existing (off-policy) DRL algorithms.

We use BEBU to propagate the future uncertainty in an episode, which is an extension of EBU (Lee et al., 2019). Compared to EBU, BEBU requires extra tensors to store the UCB-bonus for immediate reward and next-$Q$ value, which are integrated to propagate uncertainties. Meanwhile, integrating uncertainty into BEBU needs special design by using the mask. The previous works do not propagate the future uncertainty and, therefore, does not capture the core benefit of utilizing UCB-bonus for the exploration of MDPs. We highlight that OEB3 propagates future uncertainty in a time-consistent manner based on BEBU, which exploits the theoretical analysis established by Jin et al. (2020). The backward update significantly improves the sample-efficiency by allowing bonuses and delayed rewards to propagate through transitions of the sampled episode, which we demonstrate in the sequel.

## 4 RELATED WORK

One practical principle for exploration in DRL is maintaining epistemic uncertainty of action-value functions and learning to reduce the uncertainty. Epistemic uncertainty appears due to the missing

knowledge of the environment, and disappears with the progress of exploration. Bootstrapped DQN (Osband et al., 2016; 2018) samples $Q$-values from the randomized value function to encourage exploration through Thompson sampling. Chen et al. (2017) proposes to use the standard-deviation of bootstrapped Q-functions to measure the uncertainty. While the uncertainty measurement is similar to that of OEB3, our method is different from Chen et al. (2017) in the following aspects. First, our approach propagates the uncertainty through time by the backward update, which allows for deep exploration for the MDPs. Second, Chen et al. (2017) does not use the bonus in the update of $Q$-functions. The bonus is computed when taking the actions. Third, we establish a theoretical connection between the UCB-bonus and the bonus in optimistic-LSVI. SUNRISE (Lee et al., 2020) extends Chen et al. (2017) to continuous control through confidence reward and weighted Bellman backup. Information-Directed Sampling (IDS) (Nikolov et al., 2019) is based on bootstrapped DQN, and chooses actions by balancing the instantaneous regret and information gain. OAC (Ciosek et al., 2019) uses two $Q$-networks to get lower and upper bounds of $Q$-value to perform exploration in continuous control tasks. These methods seek to estimate the epistemic uncertainty and choose the optimistic actions. In contrast, we use the uncertainty of value function to construct intrinsic rewards and perform backward update, which propagates future uncertainty to the estimated $Q$-value.

Uncertainty Bellman Equation (UBE) and Exploration (O'Donoghue et al., 2018) proposes an upper bound on the variance of the posterior of the $Q$-values, which are further utilized for optimism in exploration. UBE uses posterior sampling for exploration, whereas OEB3 uses optimism for exploration. Bayesian-DQN (Azizzadenesheli et al., 2018) modifies the linear regression of the last layer in $Q$-network and uses Bayesian Linear Regression (BLR) instead, which estimates an posterior of the $Q$-function. These methods use parametric distributions to describe the posterior while OEB3 uses a non-parametric method to estimate the confidence bonus. UBE and BLR require inverting a large matrix in training. Previous methods also utilize the epistemic uncertainty of dynamics through Bayesian posterior (Ratzlaff et al., 2020) and ensembles (Pathak et al., 2019). Nevertheless, these works consider single-step uncertainty, while we consider the long-term uncertainty in an episode.

To measure the novelty of states for constructing count-based intrinsic rewards, previous methods use density model (Bellemare et al., 2016; Ostrovski et al., 2017), static hashing (Tang et al., 2017; Choi et al., 2019; Rashid et al., 2020), episodic curiosity (Savinov et al., 2019; Badia et al., 2020), representation changes (Raileanu & Rocktäschel, 2020), curiosity-bottleneck (Kim et al., 2019b), information gain (Houthooft et al., 2016) and RND (Burda et al., 2019b). The curiosity-driven exploration based on prediction-error of environment models such as ICM (Pathak et al., 2017; Burda et al., 2019a), EMI (Kim et al., 2019a), variational dynamics (Bai et al., 2020) and learning progress (Kim et al., 2020) enable the agents to explore in a self-supervised manner. According Taiga et al. (2020), although bonus-based methods show promising results in hard exploration tasks like Montezuma's Revenge, they do not perform well on other Atari games. Meanwhile, NoisyNet (Fortunato et al., 2018) performs significantly better than popular bonus-based methods evaluated by the whole Atari suit. Taiga et al. (2020) suggests that the real pace of progress in exploration may have been obfuscated by good results on several hard exploration games. We follow this principle and evaluate OEB3 on the whole Atari suit with 49 games.

Beyond model-free methods, model-based RL also uses optimism for planning and exploration (Nix & Weigend, 1994). Model-assisted RL (Kalweit & Boedecker, 2017) uses ensembles to make use of artificial data only in cases of high uncertainty. Buckman et al. (2018) uses ensemble dynamics and Q-functions to use model rollouts when they do not cause large errors. Planning to explore (Sekar et al., 2020) seeks out future uncertainty by integrating uncertainty to Dreamer (Hafner et al., 2020). Ready Policy One (Ball et al., 2020) optimizes policies for both reward and model uncertainty reduction. Noise-Augmented RL (Pacchiano et al., 2020) uses statistical bootstrap to generalize the optimistic posterior sampling (Agrawal & Jia, 2017) to DRL. Hallucinated UCRL (Curi et al., 2020) reduces optimistic exploration to exploitation by enlarging the control space. The model-based RL needs to estimate the posterior of dynamics, while OEB3 relies on the posterior of $Q$-functions.

## 5 EXPERIMENTAL RESULTS

### 5.1 ENVIRONMENTAL SETUP

We evaluate the algorithms in high-dimensional image-based tasks, including MNIST Maze and 49 Atari games. We refer Appendix C for the experiments on MNIST Maze, and discuss the experiments

on Atari games in the sequel. Comparing OEB3 with bootstrapped DQN-based baselines directly incur performance bias, since OEB3 uses backward update for training, which is different from bootstrapped DQN. To this end, we re-implement all bootstrapped DQN-based baselines with BEBU-based update. In our experiments, we compare the following methods: (1) **OEB3**. The proposed optimistic exploration method with UCB-bonus and backward update in this work. (2) **BEBU**. The re-implementation of bootstrapped DQN (Osband et al., 2016) with BEBU-based training. (3) **BEBU-UCB**. BEBU with optimistic action-selection according to the upper bound of $Q$ (Chen et al., 2017). (4) **BEBU-IDS**. Integrating homoscedastic IDS (Nikolov et al., 2019) into BEBU without distributional RL. We refer to Appendix B for the algorithmic comparison between all methods.

According to EBU (Lee et al., 2019), the backward update is significantly more sample-efficient than ordinary $Q$-learning by using 20M training frames to achieve the mean human-normalized score of DQN with 200M training frames. We follow this setting by training all BEBU-based methods with 20M frames. In our experiments, 20M frames in OEB3 is sufficient to produce strong empirical results and achieve competitive results with several baselines in 200M frames. After training, an ensemble policy by a majority vote of $Q$-heads is used for 30 no-op evaluation. The majority-vote combines all the heads into a single ensemble policy, which follows the same evaluation method as in Osband et al. (2016). We use the popular human-normalized score $\frac{\text{Score}_{\text{Agent}} - \text{Score}_{\text{Random}}}{|\text{Score}_{\text{human}} - \text{Score}_{\text{random}}|}$ to perform the comparison. In tackling Atari games, Osband et al. (2016) observes that the bootstrapping does not contribute much in performance. Empirically, Bootstrapped DQN uses the same samples to train all $Q$-heads in each training step. This empirical simplification is also adopted by Chen et al. (2017); Osband et al. (2018); Nikolov et al. (2019). We use such an simplification for OEB3 and all Bootstrapped DQN-based baselines.

We set $\alpha_1$ and $\alpha_2$ to the same value of $0.5 \times 10^{-4}$ by searching coarsely. We use diffusion factor $\beta = 0.5$ for all methods by following Lee et al. (2019). OEB3 requires much less training time than BEBU-UCB and BEBU-IDS because the confidence bound used in BEBU-UCB and regret-information ratio used in BEBU-IDS are both computed in each time step when interacting with the environment, while the UCB-bonuses in OEB3 are calculated in episodes when performing batch training. The number of environmental interaction steps $L_1$ are typically set to be much larger than the number of training steps $L_2$ (e.g., in DQN, $L_1 \approx 4L_2$). We refer to Appendix D for the detailed specifications. The code is available at https://bit.ly/33jv1ab.

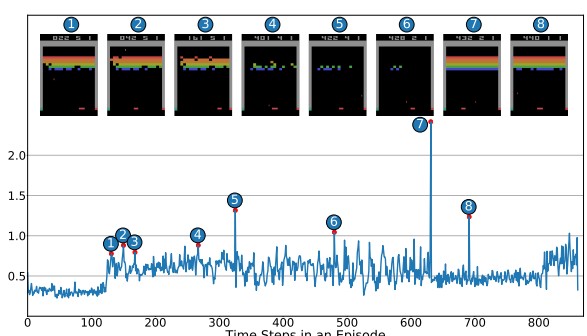

Figure 2: Visualizing UCB-bonus in Breakout

## 5.2 RESULT COMPARISON

In Table 1, we additionally report the performance of DQN (Mnih et al., 2015), NoisyNet (Fortunato et al., 2018), Bootstrapped DQN (Osband et al., 2016), and IDS (Nikolov et al., 2019) in 200M training frames. We choose **NoisyNet** as a baseline since it is shown (Taiga et al., 2020) to perform substantially better than existing bonus-based methods evaluated by the whole Atari suit (instead of several hard exploration games), including CTS-counts (Bellemare et al., 2016), PixelCNN-counts (Ostrovski et al., 2017), RND (Burda et al., 2019b), and ICM (Pathak et al., 2017). **UBE** (O'Donoghue

Table 1: Summary of human-normalized scores in 49 Atari games. BEBU, BEBU-UCB, BEBU-IDS and OEB3 are trained for 20M frames with RTX-2080Ti GPU for 5 random seeds.

| Frames | 200M | | | | | 20M | | | |
|---|---|---|---|---|---|---|---|---|---|
| | DQN | UBE | BootDQN | NoisyNet | **BootDQN-IDS** | BEBU | BEBU-UCB | **BEBU-IDS** | **OEB3** |
| Mean | 241% | 440% | 553% | 651% | **757%** | 553% | 610% | **622%** | **765%** |
| Median | 93% | 126% | 139% | 172% | **187%** | 36% | 38% | **44%** | **50%** |

et al., 2018) uses a parametric method to describe the posterior of $Q$-values, which are utilized for optimism in exploration. We also use UBE as a baseline. According to Table 1, BootDQN-IDS performs better than UBE, BootDQN, and NoisyNet. Thus, BootDQN-IDS outperforms existing bonus-based exploration methods that perform worse than NoisyNet. We re-implement BootDQN-IDS with BEBU-based update and observe that OEB3 outperforms BEBU-IDS in both mean and medium scores, thus outperforming the bonus-based methods that performs worse than NoisyNet. We report the raw scores in Appendix F. Moreover, Appendix G shows that OEB3 outperforms BEBU, BEBU-UCB, and BEBU-IDS in 36, 34, and 35 games out of all 49 games, respectively.

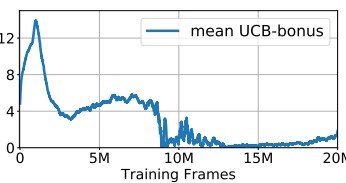

Figure 3: The change of mean UCB-bonus in Breakout

To understand the proposed UCB-bonus, we use a trained OEB3 agent to interact with the environment for an episode in Breakout and record the UCB-bonuses at each step. The curve in Figure 2 shows the UCB-bonuses of the subsampled steps in the episode. We choose 8 spikes with high UCB-bonuses and visualize the corresponding frames. The events in spikes correspond to scarcely visited areas or crucial events, which are important for the agent to obtain rewards: digging a tunnel (1), throwing the ball to the top of bricks (2,3), rebounding the ball (4,5,6), eliminating all bricks and starting a new round (7,8). We provide more examples of visualization in Appendix E. We further record the the mean of UCB-bonus of the training batch in the learning process, which is shown in Figure 3. The UCB-bonus is low at the beginning since the networks are randomly initialized. When the agent starts to explore the environment, the mean UCB-bonus rises rapidly to incentive exploration. As more experiences of states and actions are gathered, the mean UCB-bonus reduces gradually, which indicates that the bootstrapped value functions concentrate around the optimal value and the epistemic uncertainty decreases. Nevertheless, the UCB-bonuses are relatively high at scarcely visited areas or crucial events, according to Figure 2, which motivates the agent to enhance exploration at the corresponding events.

We conduct an ablation study to better comprehend the importance of backward update and bonus term in OEB3. We refer to Table 2 for the outcomes. We observe that: (1) when we use the ordinary update strategy by sampling transitions instead of episodes in train-

Table 2: Ablation Study

|  | Backward | Bonus | Qbert | SpaceInvaders | Freeway |
|---|---|---|---|---|---|
| OEB3 | ✓ | UCB | 4275.0 | 904.9 | 32.1 |
| BootDQN-UCB | - | UCB | 3284.7 | 731.8 | 20.5 |
| BEBU | ✓ | - | 3588.4 | 814.4 | 21.5 |
| BootDQN | - | - | 2206.8 | 649.5 | 18.3 |
| BEBU-RND | ✓ | RND | 3702.5 | 832.7 | 22.6 |

ing, OEB3 reduces to BootDQN-UCB with significant performance loss. Hence, the backward update is crucial in OEB3 for sample-efficient training. (2) When the UCB-bonus is set to 0, OEB3 reduces to BEBU. We observe that OEB3 outperforms BEBU in 36 out of all 49 games. (3) When both the backward update and UCB-bonus are removed, OEB3 reduces to standard BootDQN, which performs poorly in 20M training frames. (4) To illustrate the effect of UCB-bonus, we substitute UCB-bonus with the popular RND-bonus (Burda et al., 2019b). Specifically, we use an independent RND network to generate RND-bonus for each state in training. The RND-bonus is added to both the immediate reward and next-$Q$. The result shows that UCB-bonus outperforms RND-bonus without introducing additional modules compared to BootDQN.

## 6 CONCLUSION

In this work, we propose OEB3, which has theoretical underpinnings from optimistic LSVI. We propose a UCB-bonus to capture the epistemic uncertainty of $Q$-function and an BEBU algorithm for sample-efficient training. We demonstrate OEB3 empirically by solving MNIST maze and Atari games and show that OEB3 outperforms several strong baselines. The visualizations suggest that high UCB-bonus corresponds to informative experiences for exploration. As far as we are concerned, our work seems to establish the first empirical attempt of uncertainty propagation in deep RL. Moreover, we observe that such a connection between theoretical analysis and practical algorithm provides relatively strong empirical performance in Atari games and outperforms several strong baselines, which hopefully gives useful insights on combining theory and practice to the community.

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

# A    UCB BONUS IN OEB3

Recall that we consider the following regularized least-square problem,

$$w_t \leftarrow \arg\min_{w \in \mathbb{R}^d} \sum_{\tau=0}^m \big[r_t(s_t^\tau, a_t^\tau) + \max_{a \in \mathcal{A}} Q_{t+1}(s_{t+1}^\tau, a) - w^\top \phi(s_t^\tau, a_t^\tau)\big]^2 + \lambda \|w\|^2. \tag{8}$$

In the sequel, we consider a Bayesian linear regression perspective of (8) that captures the intuition behind the UCB-bonus in OEB3. Our objective is to approximate the action-value function $Q_t$ via fitting the parameter $w$, such that

$$w^\top \phi(s_t, a_t) \approx r_t(s_t, a_t) + \max_{a \in \mathcal{A}} Q_{t+1}(s_{t+1}, a),$$

where $Q_{t+1}$ is given. We assume that we are given a Gaussian prior of the initial parameter $w \sim \mathcal{N}(0, \mathbf{I}/\lambda)$. With a slight abuse of notation, we denote by $w_t$ the Bayesian posterior of the parameter $w$ given the set of independent observations $\mathcal{D}_m = \{(s_t^\tau, a_t^\tau, s_{t+1}^\tau)\}_{\tau \in [0,m]}$. We further define the following noise with respect to the least-square problem in (8),

$$\epsilon = r_t(s_t, a_t) + \max_{a \in \mathcal{A}} Q_{t+1}(s_{t+1}, a) - w^\top \phi(s_t, a_t), \tag{9}$$

where $(s_t, a_t, s_{t+1})$ follows the distribution of trajectory. The following theorem justifies the UCB-bonus in OEB3 under the Bayesian linear regression perspective.

**Theorem 2** (Formal Version of Theorem 1). *We assume that $\epsilon$ follows the standard Gaussian distribution $\mathcal{N}(0, 1)$ given the state-action pair $(s_t, a_t)$ and the parameter $w$. Let $w$ follows the Gaussian prior $\mathcal{N}(0, \mathbf{I}/\lambda)$. We define*

$$\Lambda_t = \sum_{\tau=0}^m \phi(x_t^\tau, a_t^\tau)\phi(x_t^\tau, a_t^\tau)^\top + \lambda \cdot \mathbf{I}. \tag{10}$$

*It then holds for the posterior of $w_t$ given the set of independent observations $\mathcal{D}_m = \{(s_t^\tau, a_t^\tau, s_{t+1}^\tau)\}_{\tau \in [0,m]}$ that*

$$\mathrm{Var}\big(\phi(s_t, a_t)^\top w_t\big) = \mathrm{Var}(\tilde{Q}_t(s_t, a_t)) = \phi(s_t, a_t)^\top \Lambda_t^{-1} \phi(s_t, a_t), \quad \forall (s_t, a_t) \in \mathcal{S} \times \mathcal{A}.$$

*Here we denote by $\tilde{Q}_t = w_t^\top \phi$ the estimated action-value function.*

*Proof.* The proof follows the standard analysis of Bayesian linear regression. See, e.g., West (1984) for a detailed analysis. We denote the target of the linear regression in (8) by

$$y_t = r_t(s_t, a_t) + \max_{a \in \mathcal{A}} Q_{t+1}(s_{t+1}, a).$$

By the assumption that $\epsilon$ follows the standard Gaussian distribution, we obtain that

$$y_t \,|\, (s_t, a_t), w \sim \mathcal{N}\big(w^\top \phi(s_t, a_t), 1\big). \tag{11}$$

Recall that we have the prior distribution $w \sim \mathcal{N}(0, \mathbf{I}/\lambda)$. Our objective is to compute the posterior density $w_t = w \,|\, \mathcal{D}_m$, where $\mathcal{D}_m = \{(s_t^\tau, a_t^\tau, s_{t+1}^\tau)\}_{\tau \in [0,m]}$ is the set of observations. It holds from Bayes rule that

$$\log p(w \,|\, \mathcal{D}_m) = \log p(w) + \log p(\mathcal{D}_m \,|\, w) + Const., \tag{12}$$

where $p(\cdot)$ denote the probability density function of the respective distributions. Plugging (11) and the probability density function of Gaussian distribution into (12) yields

$$\log p(w \,|\, \mathcal{D}_m) = -\|w\|^2/2 - \sum_{\tau=1}^m \|w^\top \phi(s_t^\tau, a_t^\tau) - y_t^\tau\|^2/2 + Const.$$

$$= -(w - \mu_t)^\top \Lambda_t^{-1}(w - \mu_t)/2 + Const., \tag{13}$$

where we define

$$\mu_t = \Lambda_t^{-1} \sum_{\tau=1}^m \phi(s_t^\tau, a_t^\tau)y_t^\tau, \qquad \Lambda_t = \sum_{\tau=0}^m \phi(x_t^\tau, a_t^\tau)\phi(x_t^\tau, a_t^\tau)^\top + \lambda \cdot \mathbf{I}.$$

Thus, by (13), we obtain that $w_t = w \,|\, \mathcal{D}_m \sim \mathcal{N}(\mu_t, \Lambda_t^{-1})$. It then holds for all $(s_t, a_t) \in \mathcal{S} \times \mathcal{A}$ that

$$\mathrm{Var}\big(\phi(s_t, a_t)^\top w_t\big) = \mathrm{Var}\big(\tilde{Q}_t(s_t, a_t)\big) = \phi(s_t, a_t)^\top \Lambda_t^{-1} \phi(s_t, a_t),$$

which concludes the proof of Theorem 2. □

# B   ALGORITHMIC DESCRIPTION

---

**Algorithm 2** OEB3 in DRL

---

1: **Initialize:** replay buffer $\mathcal{D}$, bootstrapped $Q$-network $Q(\cdot;\theta)$ and target network $Q(\cdot;\theta^-)$
2: **Initialize:** total training frames $H = 20$M, current frame $h = 0$
3: **while** $h < H$ **do**
4:   Pick a bootstrapped $Q$-function to act by sampling $k \sim \text{Unif}\{1,\dots,K\}$
5:   Reset the environment and receive the initial state $s_0$
6:   **for** step $i = 0$ **to** Terminal **do**
7:     With $\epsilon$-greedy choose $a_i = \arg\max_a Q^k(s_i, a)$
8:     Take action and observe $r_i$ and $s_{i+1}$, then save the transition in buffer $\mathcal{D}$
9:     **if** $h \% $ training frequency $= 0$ **then**
10:       Sample an episodic experience $E = \{\mathbf{S}, \mathbf{A}, \mathbf{R}, \mathbf{S'}\}$ with length $T$ from $\mathcal{D}$
11:       Initialize a $Q$-table $\tilde{\mathbf{Q}} = Q(\mathbf{S'}, \mathcal{A}; \theta^-) \in \mathbb{R}^{K \times |\mathcal{A}| \times T}$ by the target $Q$-network
12:       Compute the UCB-bonus for immediate reward for all steps to construct $\mathbf{B} \in \mathbb{R}^T$
13:       Compute the action matrix $\tilde{\mathbf{A}} = \arg\max_a \tilde{\mathbf{Q}}[\cdot, a, \cdot] \in \mathbb{R}^{K \times T}$ to gather all $a'$ of next-$Q$
14:       Compute the UCB-bonus for next-$Q$ for all heads and all steps to construct $\tilde{\mathbf{B}} \in \mathbb{R}^{K \times T}$
15:       Compute the mask matrix $\mathbf{M} \in \mathbb{R}^{K \times T}$ where $\mathbf{M}[k, t] = \mathbb{1}_{\tilde{\mathbf{A}}[k,t] \neq \mathbf{A}_{t+1}}$
16:       Initialize target table $\mathbf{y} \in \mathbb{R}^{K \times T}$ to zeros, and set $\mathbf{y}[\cdot, T-1] = \mathbf{R}_{T-1} + \alpha_1 \mathbf{B}_{T-1}$
17:       **for** $t = T - 2$ **to** $0$ **do**
18:         $\tilde{\mathbf{Q}}[\cdot, a_{t+1}, t] \leftarrow \beta \mathbf{y}[\cdot, t+1] + (1-\beta)\tilde{\mathbf{Q}}[\cdot, a_{t+1}, t]$
19:         $\mathbf{y}[\cdot, t] \leftarrow (\mathbf{R}_t + \alpha_1 \mathbf{B}_t) + \gamma(\tilde{\mathbf{Q}}[\cdot, a', t] + \alpha_2 \mathbf{M}[\cdot, t] \circ \tilde{\mathbf{B}}[\cdot, t])$ where $a' = \tilde{\mathbf{A}}[\cdot, t]$
20:       **end for**
21:       Compute the $Q$-value of $(\mathbf{S}, \mathbf{A})$ for all heads as $\mathbf{Q} = Q(\mathbf{S}, \mathbf{A}; \theta) \in \mathbb{R}^{K \times T}$
22:       Perform a gradient descent step on $(\mathbf{y} - \mathbf{Q})^2$ with respect to $\theta$
23:     **end if**
24:     Every $C$ steps reset $\theta^- \leftarrow \theta$
25:     $h \leftarrow h + 1$
26:   **end for**
27: **end while**

---

We summarize the closely related works in Table 3.

Table 3: Algorithmic Comparison of Related Works

|  | Bonus or Posterior Variance | Update Method | Uncertainty Characterization |
|---|---|---|---|
| EBU (Lee et al., 2019) | - | backward update | - |
| Bootstrapped DQN (Osband et al., 2016) | bootstrapped | on-trajectory update | bootstrapped distribution |
| BEBU (base of our work) | bootstrapped | backward update | bootstrapped distribution |
| Optimistic LSVI (Jin et al., 2020) | closed form | backward update | optimism |
| OEB3 (ours) | bootstrapped | backward update | optimism |
| UBE (O'Donoghue et al., 2018) | closed form | on-trajectory update | posterior sampling |
| Bayesian-DQN (Azizzadenesheli et al., 2018) | closed form | on-trajectory update | posterior sampling |

---

**Algorithm 3** BEBU & BEBU-UCB & BEBU-IDS

---

1: **Input:** Algorithm Type (BEBU, BEBU-UCB, or BEBU-IDS)
2: **Initialize:** replay buffer $\mathcal{D}$, bootstrapped $Q$-network $Q(\cdot; \theta)$ and target network $Q(\cdot; \theta^-)$
3: **Initialize:** total training frames $H = 20\text{M}$, current frame $h = 0$
4: **while** $h < H$ **do**
5:      Pick a bootstrapped $Q$-function to act by sampling $k \sim \text{Unif}\{1, \dots, \text{K}\}$
6:      Reset the environment and receive the initial state $s_0$
7:      **for** step $i = 0$ **to** Terminal **do**
8:          **if** Algorithm type is BEBU **then**
9:              With $\epsilon$-greedy choose $a_i = \arg\max_a Q^k(s_i, a)$
10:          **else if** Algorithm type is BEBU-UCB **then**
11:              With $\epsilon$-greedy choose $a_i = \arg\max_a[\bar{Q}(s_i, a) + \alpha \cdot \sigma(Q(s_i, a))]$, where $\bar{Q}(s_i, a_i) = \frac{1}{K}\sum_{k=1}^{K} Q^k(s_i, a_i)$ and $\sigma(Q(s_i, a_i)) = \sqrt{\frac{1}{K}\sum_{k=1}^{K}(Q^k(s_i, a_i) - \bar{Q}(s_i, a_i))^2}$ are the mean and standard deviation of the bootstrapped Q-estimates
12:          **else if** Algorithm type is BEBU-IDS **then**
13:              With $\epsilon$-greedy choose $a_i = \arg\min_a \frac{\hat{\Delta}_i(s_i, a)^2}{I_i(s_i, a)}$ by following the regret-information ratio, where $\hat{\Delta}_i(s_i, a_i) = \max_{a' \in \mathcal{A}} u_i(s_i, a') - l_i(s_i, a_i)$ is the expected regret, and $[l_i(s_i, a_i), u_i(s_i, a_i)]$ is the confidence interval. In particular, $u_i(s_i, a_i) = \bar{Q}(s_i, a_i) + \lambda_{\text{ids}} \cdot \sigma(Q(s_i, a_i))$ and $l_i(s_i, a_i) = \bar{Q}(s_i, a_i) - \lambda_{\text{ids}} \cdot \sigma(Q(s_i, a_i))$. $I(s_i, a_i) = \log(1 + \sigma(Q(s_i, a_i))^2/\rho^2) + \epsilon_{\text{ids}}$ measures the uncertainty, where $\rho$ and $\epsilon_{\text{ids}}$ are constants.
14:          **else**
15:              Algorithm type error.
16:          **end if**
17:          Take action and observe $r_i$ and $s_{i+1}$, then save the transition in buffer $\mathcal{D}$
18:          **if** $h \% \text{training frequency} = 0$ **then**
19:              Sample an episodic experience $E = \{\mathbf{S}, \mathbf{A}, \mathbf{R}, \mathbf{S}'\}$ with length $T$ from $\mathcal{D}$
20:              Initialize a $Q$-table $\tilde{\mathbf{Q}} = Q(\mathbf{S}', \mathcal{A}; \theta^-) \in \mathbb{R}^{K \times |\mathcal{A}| \times T}$ by the target $Q$-network
21:              Compute the action matrix $\tilde{\mathbf{A}} = \arg\max_a \tilde{\mathbf{Q}}[\cdot, a, \cdot] \in \mathbb{R}^{K \times T}$ to gather all $a'$ of next-$Q$
22:              Initialize target table $\mathbf{y} \in \mathbb{R}^{K \times T}$ to zeros, and set $\mathbf{y}[\cdot, T-1] = \mathbf{R}_{T-1} + \alpha_1 \mathbf{B}_{T-1}$
23:              **for** $t = T - 2$ **to** $0$ **do**
24:                  $\tilde{\mathbf{Q}}[\cdot, a_{t+1}, t] \leftarrow \beta \mathbf{y}[\cdot, t+1] + (1 - \beta)\tilde{\mathbf{Q}}[\cdot, a_{t+1}, t]$
25:                  $\mathbf{y}[\cdot, t] \leftarrow \mathbf{R}_t + \gamma \tilde{\mathbf{Q}}[\cdot, a', t]$ where $a' = \tilde{\mathbf{A}}[\cdot, t]$
26:              **end for**
27:              Compute the $Q$-value of $(\mathbf{S}, \mathbf{A})$ for all heads as $\mathbf{Q} = Q(\mathbf{S}, \mathbf{A}; \theta) \in \mathbb{R}^{K \times T}$
28:              Perform a gradient descent step on $(\mathbf{y} - \mathbf{Q})^2$ with respect to $\theta$
29:          **end if**
30:          Every $C$ steps reset $\theta^- \leftarrow \theta$
31:          $h \leftarrow h + 1$
32:      **end for**
33: **end while**

---

# C ADDITIONAL EXPERIMENT: MNIST MAZE

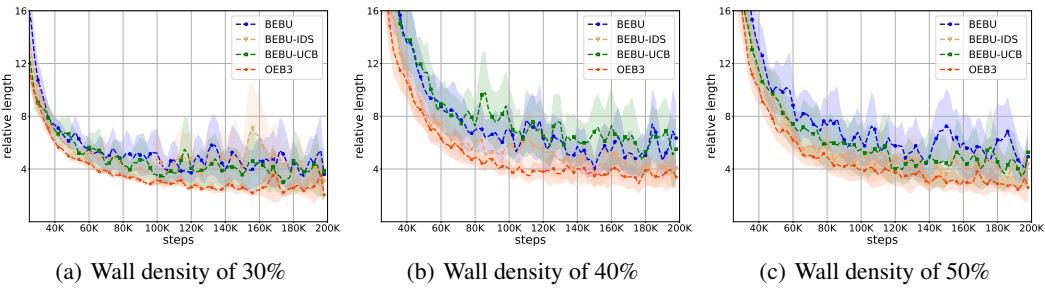

(a) Wall density of 30%    (b) Wall density of 40%    (c) Wall density of 50%

Figure 4: Results of 200K steps training of MNIST maze with different wall-density setup.

We use $10 \times 10$ MNIST maze with randomly placed walls to evaluate our method. The agent starts from the initial position $(0, 0)$ in the upper-left of the maze and aims to reach the goal position $(9, 9)$ in the bottom-right. The state of position $(i, j)$ is represented by stacking two randomly sampled images with label $i$ and $j$ from the MNIST dataset. When the agent steps to a new position, the state representation is reconstructed by sampling images. Hence the agent gets different states even stepping to the same location twice, which minimizes the correlation among locations. We further add some stochasticity in the transition probability. The agent has a probabil-

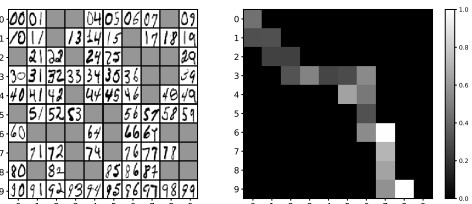

Figure 5: An example MNIST maze (left) and the UCB-bonuses in the agent's path (right).

ity of 10% to arrive in the adjacent locations when taking an action. For example, when taking action 'left', the agent has a 10% chance of transiting to 'up', and a 10% chance of transiting to 'down'. The agent gets a reward of -1 when bumping into a wall, and gets 1000 when reaching the goal.

We use the different setup of wall-density in the experiment. The wall-density means the proportion of walls in all locations. Figure 5 (left) shows a generated maze with wall-density of 50%. The gray positions represent walls. We train all methods with wall-density of 30%, 40%, and 50%. For each setup, we train 50 independent agents for 50 randomly generated mazes. The relative length defined by $l_{\text{agent}}/l_{\text{best}}$ is used to evaluate the performance, where $l_{\text{agent}}$ is the length of the agent's path in an episode (maximum steps are 1000), and $l_{\text{best}}$ is the length of the shortest path. The performance comparison is shown in Figure 4. OEB3 performs best in all methods, and BEBU-IDS is a strong baseline. We use a trained OEB3 agent to take action in the maze of Figure 5 (left). The UCB-bonuses of state-action pairs in the agent's path are computed and visualized in Figure 5 (right). The state-action pairs with high UCB-bonus show the bottleneck positions in the path. For example, the state-action pairs in location $(3, 3)$ and $(6, 7)$ produce high UCB-bonus to guide the agent to choose the right direction. The UCB-bonus encourages the agent to walk through these bottleneck positions correctly. We give more examples of visualization in Appendix E.

# D  IMPLEMENTATION DETAIL

## D.1  MNIST MAZE

**Hyper-parameters of BEBU**. BEBU is the basic algorithm of BEBU-UCB and BEBU-IDS. BEBU uses the same network-architecture as Bootstrapped DQN (Osband et al., 2016). The diffusion factor and other training parameters are set by following EBU paper (Lee et al., 2019). Details are summarized in Table 4.

Table 4: Hyper-parameters of BEBU for MNIST-Maze

| Hyperparameters | Value | Description |
|---|---|---|
| state space | $28 \times 28 \times 2$ | Stacking two images sampled from MNIST dataset with labels according to the agent's current location. |
| action space | 4 | Including left, right, up, and down. |
| $K$ | 10 | Number of bootstrapped heads. |
| network-architecture | conv(64,4,4) conv(64,3,1) dense$\{512, 4\}_{k=1}^{K}$ | Using convolution (channels, kernel size, stride) layers first, then fully connected into $K$ bootstrapped heads. Each head has 512 ReLUs and 4 linear units. |
| gradient norm | 10 | The gradient is clipped by 10. The gradient of each head is normalize by $1/K$ according to bootstrapped DQN. |
| learning starts | 10000 | The agent takes actions according to the initial policy before learning starts. |
| replay buffer size | 170 | A simple replay buffer is used to store episodic experience. |
| training frequency | 50 | Number of action-selection step between successive gradient descent steps. |
| $H$ | 200,000 | Total timesteps to train a single maze. |
| target network update frequency | 2000 | The target-network is updated every 2000 steps. |
| optimizer | Adam | Adam optimizer is used for training. Detailed parameters: $\beta_1 = 0.9$, $\beta_2 = 0.999$, $\epsilon_{\text{ADAM}} = 10^{-7}$. |
| learning rate | 0.001 | Learning rate for Adam optimizer. |
| $\epsilon$ | $\frac{(h-H)^2}{H^2}$ | Exploration factor. $H$ is the total timesteps for training, and $h$ is the current timestep. $\epsilon$ starts from 1 and is annealed to 0 in a quadratic manner. |
| $\gamma$ | 0.9 | Discount factor. |
| $\beta$ | 1.0 | Diffusion factor of backward update. |
| wall density | 30%, 40%, and 50% | Proportion of walls in all locations of the maze. |
| reward | -1 or 1000 | Reward is -1 when bumping into a wall, and 1000 when reaching the goal. |
| stochasticity | 10% | Has a probability of 10% to arrive in the adjacent locations when taking an action. |
| evaluation metric | $l_{\text{rel}} = l_{\text{agent}}/l_{\text{best}}$ | Ratio between length of the agent's path and the best length. |

**Hyper-parameters of BEBU-UCB**. BEBU-UCB uses the upper-bound of $Q$-values to select actions. In particular, $a = \arg\max_{a \in \mathcal{A}} [\mu(s, a) + \lambda_{\text{ucb}}\sigma(s, a)]$, where $\mu(s, a)$ and $\sigma(s, a)$ are the mean and standard deviation of bootstrapped $Q$-values $\{Q^k(s, a)\}_{k=1}^{K}$. We use $\lambda_{\text{ucb}} = 0.1$ in our experiment.

**Hyper-parameters of BEBU-IDS**. The action-selection in IDS (Nikolov et al., 2019) follows the regret-information ratio as $a_t = \arg\min_{a \in \mathcal{A}} \frac{\hat{\Delta}_t(s,a)^2}{I_t(s,a)}$, which balances the regret and exploration. $\hat{\Delta}_t(s, a)$ is the expected regret that indicates the loss of reward when choosing a suboptimal action $a$. IDS uses a conservative estimate of regret as $\hat{\Delta}_t(s, a) = \max_{a' \in \mathcal{A}} u_t(s, a') - l_t(s, a)$, where $[l_t(s, a), u_t(s, a)]$ is the confidence interval of action-value function. In particular, $u_t(s, a) = \mu(s, a) + \lambda_{\text{ids}}\sigma(s, a)$ and $l_t(s, a) = \mu(s, a) - \lambda_{\text{ids}}\sigma(s, a)$, where $\mu(s, a)$ and $\sigma(s, a)$ are the mean and standard deviation of bootstrapped $Q$-values $\{Q^k(s, a)\}_{k=1}^{K}$. The information gain $I_t(a)$ measures the uncertainty of action-values as $I(s, a) = \log(1 + \frac{\sigma(s,a)^2}{\rho(s,a)^2}) + \epsilon_{\text{ids}}$, where $\rho(s, a)$ is the variance of the return distribution. $\rho(s, a)$ is measured by C51 (Bellemare et al., 2017) in distributional RL and becomes a constant in ordinary $Q$-learning. We use $\lambda_{\text{ids}} = 0.1$, $\rho(s, a) = 1.0$, and $\epsilon_{\text{ids}} = 10^{-5}$ in our experiment.

**Hyper-parameters of OEB3**. We set $\alpha_1$ and $\alpha_2$ to the same value of $0.01$. We find that adding a normalizer to UCB-bonus $\tilde{\mathbf{B}}$ of the next $Q$-value enables us to get more stable performance. $\tilde{\mathbf{B}}$ is smoothed by dividing a running estimate of its standard deviation. Because the UCB-bonuses for next-$Q$ are different for each $Q$-head, the normalization is useful in most cases by making the value function have a smooth and stable update.

### D.2 ATARI GAMES

**Hyper-parameters of BEBU**. The basic setting of the Atari environment is the same as Nature DQN (Mnih et al., 2015) and EBU (Lee et al., 2019) papers. We use the different network architecture, environmental setup, exploration factor, and evaluation scheme from MNIST maze experiment. Details are summarized in Table 5.

Table 5: Hyper-parameters of BEBU for Atari games

| Hyperparameters | Value | Description |
|---|---|---|
| state space | $84 \times 84 \times 4$ | Stacking 4 recent frames as the input to network. |
| action repeat | 4 | Repeating each action 4 times. |
| $K$ | 10 | The number of bootstrapped heads. |
| network-architecture | conv(32,8,4) conv(64,4,2) conv(64,3,1) dense$\{512, |\mathcal{A}|\}_{k=1}^{K}$ | Using convolution(channels, kernel size, stride) layers first, then fully connected into $K$ bootstrapped heads. Each head has 512 ReLUs and $|\mathcal{A}|$ linear units. |
| gradient norm | 10 | The gradient is clipped by 10, and also be normalize by $1/K$ for each head by following bootstrapped DQN. |
| learning starts | 50000 | The agent takes random actions before learning starts. |
| replay buffer size | 1M | The number of recent transitions stored in the replay buffer. |
| training frequency | 4 | The number of action-selection step between successive gradient descent steps. |
| $H$ | 20M | Total frames to train an environment. |
| target network update frequency | 10000 | The target-network is updated every 10000 steps. |
| optimizer | Adam | Adam optimizer is used for training. Detailed parameters: $\beta_1 = 0.9$, $\beta_2 = 0.999$, $\epsilon_{\text{ADAM}} = 10^{-7}$. |
| mini-batch size | 32 | The number of training cases for gradient decent each time. |
| learning rate | 0.00025 | Learning rate for Adam optimizer. |
| initial exploration | 1.0 | Initial value of $\epsilon$ in $\epsilon$-greedy exploration. |
| final exploration | 0.1 | Final value of $\epsilon$ in $\epsilon$-greedy exploration. |
| final exploration frames | 1M | The number of frames that the initial value of $\epsilon$ linearly annealed to the final value. |
| $\gamma$ | 0.99 | Discount factor. |
| $\beta$ | 0.5 | Diffusion factor of backward update. |
| $\epsilon_{eval}$ | 0.05 | Exploration factor in $\epsilon$-greedy for evaluation. |
| evaluation policy | ensemble vote | The same evaluation method as in Bootstrapped DQN Osband et al. (2016). |
| evaluation length | 108000 | The policy is evaluated for 108000 steps. |
| evaluation frequency | 100K | The policy is evaluated every 100K steps. |
| max no-ops | 30 | Maximum number no-op actions before an episode starts. |

**Hyper-parameters of BEBU-UCB**. BEBU-UCB selects actions by $a = \arg\max_{a \in \mathcal{A}} \left[ \mu(s,a) + \lambda_{\text{ucb}} \sigma(s,a) \right]$. The detail is given in Appendix D.1. We use $\lambda_{\text{ucb}} = 0.1$ in our experiment by searching coarsely.

**Hyper-parameters of BEBU-IDS**. The action-selection follows the regret-information ratio as $a_t = \arg\min_{a \in \mathcal{A}} \frac{\hat{\Delta}_t(s,a)^2}{I_t(s,a)}$. The details are given in Appendix D.1. We use $\lambda_{\text{ids}} = 0.1$, $\rho(s,a) = 1.0$ and $\epsilon_{\text{ids}} = 10^{-5}$ in our experiment by searching coarsely.

**Hyper-parameters of OEB3**. We set $\alpha_1$ and $\alpha_2$ to the same value of $0.5 \times 10^{-4}$. The UCB-bonus $\tilde{\mathbf{B}}$ for the next $Q$-value is normalized by dividing a running estimate of its standard deviation to have a stable performance.

# E  VISUALIZING OEB3

OEB3 uses the UCB-bonus that indicates the disagreement of bootstrapped $Q$-estimates to measure the uncertainty of $Q$-functions. The state-action pairs with high UCB-bonuses signify the bottleneck positions or meaningful events. We provide visualization in several tasks to illustrate the effect of UCB-bonuses. Specifically, we choose *Mnist-maze* and two popular Atari games *Breakout* and *Mspacman* to analyze.

## E.1  MNIST-MAZE

Figure 6 illustrates the UCB-bonus in four randomly generated mazes. The mazes in Figure 6(a) and 6(b) have a wall-density of $40\%$, and in Figure 6(c) and 6(d) have a wall-density of $50\%$. The left of each figure shows the map of maze, where the black blocks represent the walls. We do not show the MNIST representation of states for simplification, and MNIST states are used in training. A trained OEB3 agent starts at the upper-left, then takes actions to achieve the goal at bottom-right. The UCB-bonuses of state-action pairs in the agent's path are computed and shown on the right of each figure. The value is normalized to $0 \sim 1$ for visualization. We show the maximal value if the agent appears several times in the same location.

The positions with UCB-bonuses that higher than 0 draw the path of the agent. The path is usually winding and includes positions beyond the shortest path because the state transition has stochasticity. The state-action pairs with high UCB-bonuses show the bottleneck positions in the path. In maze 6(a), the agent slips from the right path in position $(4, 7)$ to $(4, 9)$. The state-action in position $(4, 8)$ produces high bonus to guide the agent back to the right path. In maze 6(b), the bottleneck state in $(3, 2)$ has high bonus to avoid the agent from entering into the wrong side of the fork. The other two mazes also have bottleneck positions, like $(3, 3)$ in maze 6(c) and $(7, 6)$ in maze 6(d). Selecting actions incorrectly in these important locations is prone to failure. The UCB-bonus encourages the agent to walk through these bottleneck positions correctly.

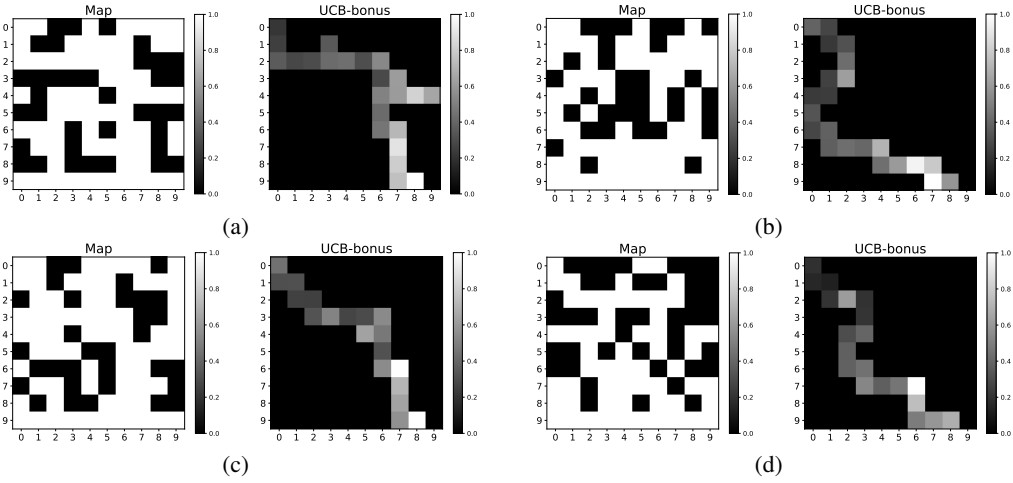

Figure 6: Visualization of UCB-bonus in Mnist-maze

## E.2  BREAKOUT

In Breakout, the agent uses walls and the paddle to rebound the ball against the bricks and eliminate them. We use a trained OEB3 agent to interact with the environment for an episode in Breakout. The whole episode contains 3445 steps, and we subsample them every 4 steps for visualization. The curve in Figure 7 shows the UCB-bonus in 861 sampled steps. We choose 16 spikes with high UCB-bonuses and visualize the corresponding frames. The events in spikes usually mean meaningful experiences that are important for the agent to get rewards. In step 1, the agent is hoping to dig a tunnel and get rewards faster. After digging a tunnel, the balls appear on the top of bricks in steps 2, 3, 4, 5, 6, 9, and 12. Balls on the top are easier to hit bricks. The agents rebound the ball and throw it over the bricks in steps 7, 8, 10, and 11. In state 13, the agent eliminates all bricks and then comes to a new

round, which is novel and promising to get more rewards. The agents in steps 14, 15, and 16 rebound the ball and try to dig a tunnel again. The UCB-bonus encourages the agent to explore the potentially informative and novel state-action pairs to get high rewards. We record 15 frames after each spike for further visualization. The video is available at `https://youtu.be/VptBkHyMt8g`.

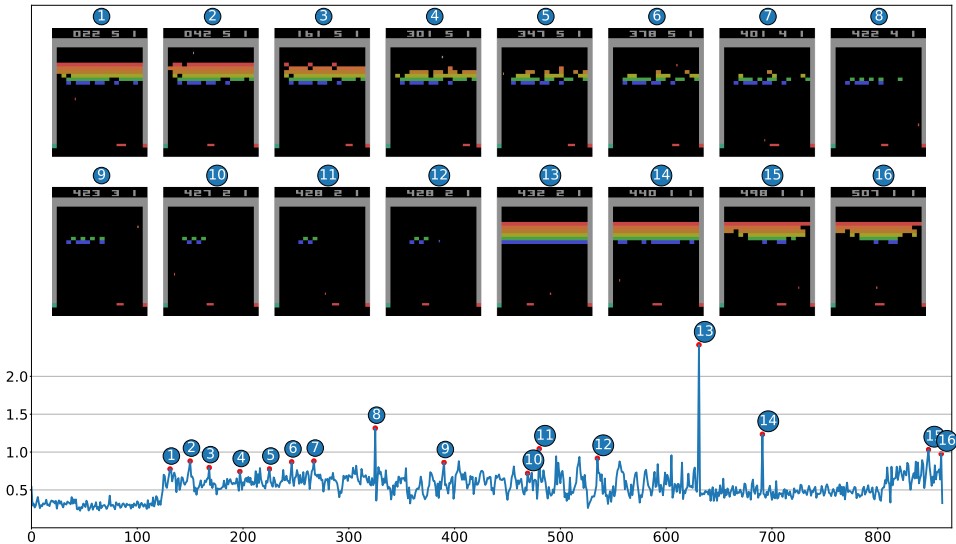

Figure 7: Visualization of UCB-bonus in Breakout

## E.3 MSPACMAN

In MsPacman, the agent earns points by avoiding monsters and eating pellets. Eating an energizer causes the monsters to turn blue, allowing them to be eaten for extra points. We use a trained OEB3 agent to interact with the environment for an episode. Figure 8 shows the UCB bonus in all 708 steps. We choose 16 spikes to visualize the frames. The spikes of exploration bonuses correspond to meaningful events for the agent to get rewards: starting a new scenario (1,2,9,10), changing direction (3,4,13,14,16), eating energizer (5,11), eating monsters (7,8,12), and entering the corner (6,15). These state-action pairs with high UCB-bonuses make the agent explore the environment efficiently. We record 15 frames after each spike, and the video is shown at `https://youtu.be/C_8NHKpBNXM`.

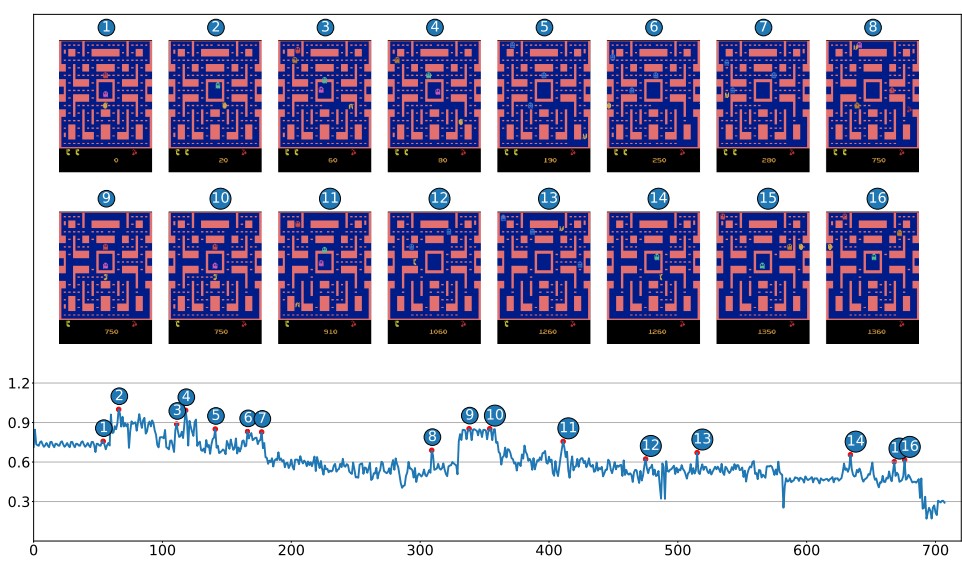

Figure 8: Visualization of UCB-bonus in MsPacman

# F    RAW SCORES OF ALL 49 ATARI GAMES

Table 6: Raw scores for Atari games. Each game is trained for 20M frames with a single RTX-2080Ti GPU. Bold scores signify the best score out of all methods.

| | Random | Human | BEBU | BEBU-UCB | BEBU-IDS | OEB3 |
|---|---|---|---|---|---|---|
| Alien | 227.8 | 6,875.0 | **1,118.0** | 811.1 | 857.9 | 916.9 |
| Amidar | 5.8 | 1676.0 | 81.7 | **166.4** | 148.1 | 94.0 |
| Assault | 222.4 | 1,496.0 | 1,377.0 | **3,574.5** | 2,441.8 | 2,996.2 |
| Asterix | 210.0 | 8,503.0 | 2,315.0 | 2,709.3 | 2,433.9 | **2,719.0** |
| Asteroids | 719.1 | 13,157.0 | 962.8 | **1,025.0** | 868.8 | 959.9 |
| Atlantis | 12,850.0 | 29,028.0 | 3,020,500.0 | 3,191,600.0 | 3,144,440.0 | **3,146,300.0** |
| Bank Heist | 14.2 | 734.4 | 331.8 | 277.0 | 361.6 | **378.6** |
| Battle Zone | 2,360.0 | 37,800.0 | 5,446.4 | **16,348.8** | 10,520.0 | 13,454.5 |
| BeamRider | 363.9 | 5,775.0 | 2,930.0 | 3,208.3 | 3,391.0 | **3,736.7** |
| Bowling | 23.1 | 154.8 | 29.9 | 30.7 | **40.2** | 30.0 |
| Boxing | 0.1 | 4.3 | 72.4 | 68.3 | 69.8 | **75.1** |
| Breakout | 1.7 | 31.8 | **473.2** | 382.3 | 412.7 | 423.1 |
| Centipede | 2,090.9 | 11,963.0 | 2,547.2 | 2,377.9 | **3,328.4** | 2,661.8 |
| Chopper Command | 811.0 | 9,882.0 | 930.6 | 1,013.4 | 1,100.0 | **1,100.3** |
| Crazy Climber | 10,780.5 | 35,411.0 | 49,735.7 | 39,187.5 | 42,242.9 | **53,346.7** |
| Demon Attack | 152.1 | 3,401.0 | 6,506.3 | 6,840.4 | **7,080.0** | 6,794.6 |
| Double Dunk | -18.6 | -15.5 | -18.9 | **-16.5** | -17.0 | -18.2 |
| Enduro | 0.0 | 309.6 | 504.1 | 697.8 | 513.6 | **719.0** |
| Fishing Derby | -91.7 | 5.5 | -56.7 | -83.8 | **-53.3** | -60.1 |
| Freeway | 0.0 | 29.6 | 21.5 | 21.6 | 21.3 | **32.1** |
| Frostbite | 65.2 | 4,335.0 | 393.4 | 470.4 | 466.2 | **1,277.3** |
| Gopher | 257.6 | 2,321.0 | 4,842.6 | **7,211.8** | 7,171.5 | 6,359.5 |
| Gravitar | 173.0 | 2,672.0 | 256.1 | 321.0 | 283.3 | **393.6** |
| H.E.R.O | 1,027.0 | 25,763.0 | 2,951.4 | 2,905.0 | 3,059.4 | **3,302.5** |
| Ice Hockey | -11.2 | 0.9 | -5.4 | -6.5 | -4.6 | **-4.2** |
| Jamesbond | 29.0 | 406.7 | **650.0** | 360.3 | 302.1 | 434.3 |
| Kangaroo | 52.0 | 3,035.0 | 3624.2 | 2,711.1 | **4,448.0** | 2,387.0 |
| Krull | 1,598.0 | 2,395.0 | 15,716.7 | 11,499.0 | 10,818.0 | **45,388.8** |
| Kung-Fu Master | 258.5 | 22,736.0 | 56.0 | 20,738.9 | **26,909.7** | 16,272.2 |
| Montezuma's Revenge | 0.0 | 4,376.0 | 0.0 | 0.0 | 0.0 | **0.0** |
| Ms. Pacman | 307.3 | 15,693.0 | 1,723.8 | 1,706.8 | 1,615.5 | **1,794.9** |
| Name This Game | 2,292.3 | 4,076.0 | 8,275.3 | 6,573.9 | **8,925.0** | 8,576.8 |
| Pong | -20.7 | 9.3 | 18.1 | 18.5 | 17.2 | **18.7** |
| Private Eye | 24.9 | 69,571.0 | 1,185.8 | 1,925.2 | 1,897.1 | 1,174.1 |
| Q*Bert | 163.9 | 13,455.0 | 3,588.4 | 3,783.2 | 3,696.0 | **4,275.0** |
| River Raid | 1,338.5 | 13,513.0 | 3,127.5 | **3,617.7** | 3,169.1 | 2,926.5 |
| Road Runner | 11.5 | 7,845.0 | 11,483.0 | 20,990.7 | 17,281.4 | **21,831.4** |
| Robotank | 2.2 | 11.9 | 10.3 | 13.3 | 10.7 | **13.5** |
| Seaquest | 68.4 | 20,182.0 | 447.0 | **492.3** | 332.4 | 332.1 |
| Space Invaders | 148.0 | 1,652.0 | 814.4 | 782.2 | 794.7 | **904.9** |
| Star Gunner | 664.0 | 10,250.0 | 1,467.2 | 1,201.5 | 1,158.9 | **1,290.2** |
| Tennis | -23.8 | -8.9 | -1.0 | -2.0 | -1.0 | **-1.0** |
| Time Pilot | 3,568.0 | 5,925.0 | 2,622.1 | 3,321.2 | 1,950.6 | **3,404.5** |
| Tutankham | 11.4 | 167.6 | 167.0 | 151.0 | 80.5 | **297.0** |
| Up and Down | 533.4 | 9,082.0 | **5,954.8** | 4,530.2 | 4,619.7 | 5,100.8 |
| Venture | 0.0 | 1,188.0 | 42.9 | 3.4 | **150.0** | 16.1 |
| Video Pinball | 16,256.9 | 17,298.0 | 26,829.6 | 48,959.1 | 58,398.3 | **80,607.0** |
| Wizard of Wor | 563.5 | 4,757.0 | 810.8 | **1,316.7** | 578.2 | 480.7 |
| Zaxxon | 32.5 | 9,173.0 | 1,587.5 | 2,104.8 | 1,594.2 | **2,842.0** |

# G   PERFORMANCE COMPARISON

We use the relative scores as

$$\frac{\text{Score}_{\text{Agent}} - \text{Score}_{\text{Baseline}}}{\max\{\text{Score}_{\text{human}}, \text{Score}_{\text{baseline}}\} - \text{Score}_{\text{random}}}$$

to compare OEB3 with baselines. The results of OEB3 comparing with BEBU, BEBU-UCB, and BEBU-IDS is shown in Figure 9, Figure 10, and Figure 11, respectively.

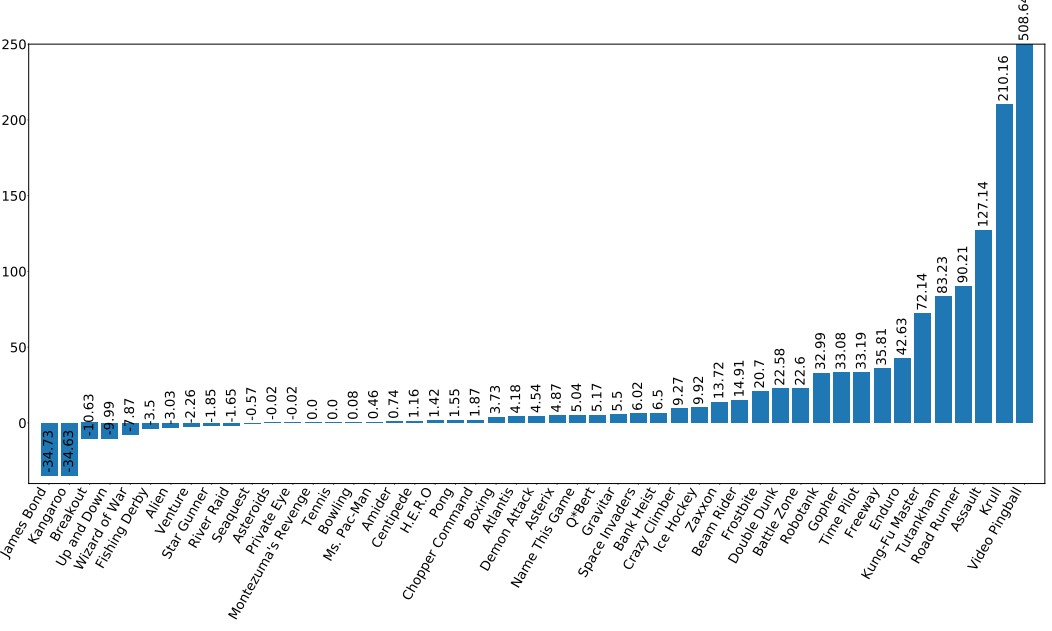

Figure 9: Relative score of OEB3 compared to BEBU in percents (%).

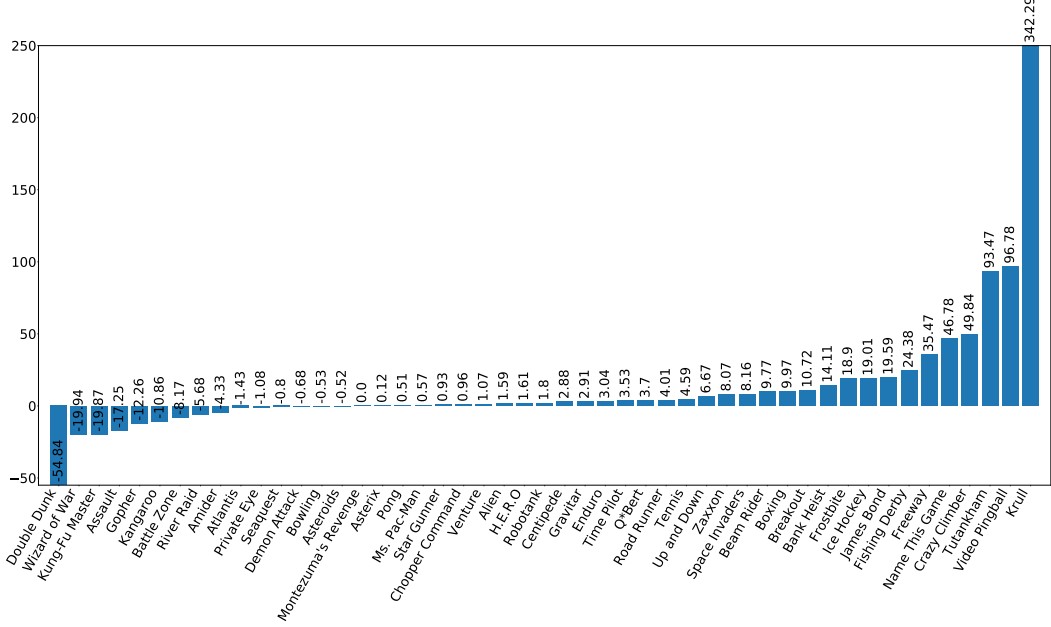

Figure 10: Relative score of OEB3 compared to BEBU-UCB in percents (%).

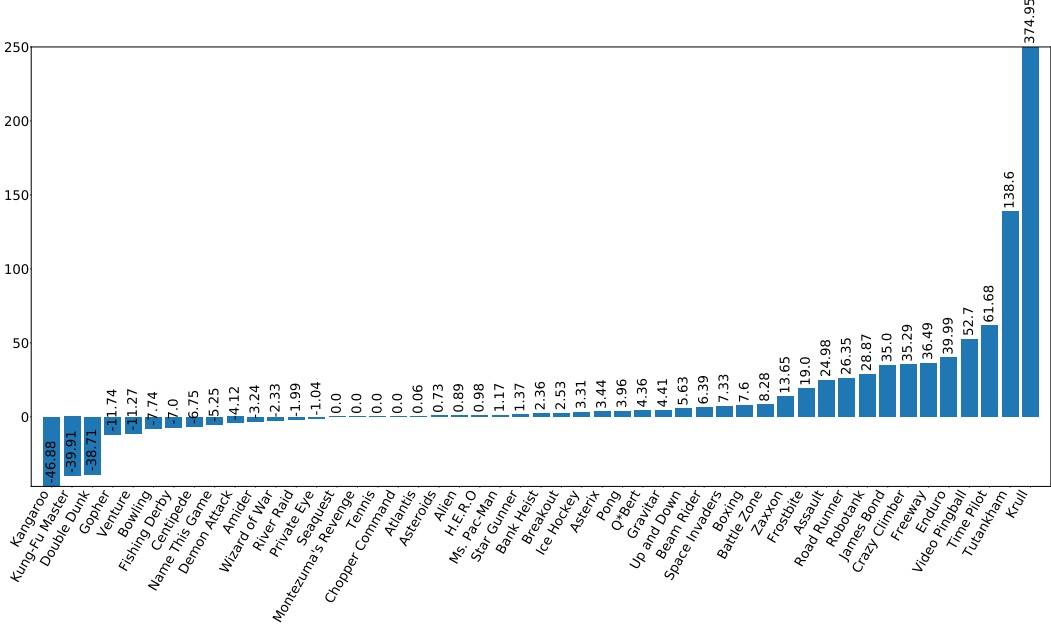

Figure 11: Relative score of OEB3 compared to BEBU-IDS in percents (%).

## H  FAILURE ANALYSIS

Table 7: Comparison of scores in *Montezuma's Revenge.*

| Frames | 200M | | | | 20M | | | |
|--------|------|---------|---------|------------|------|----------|----------|------|
| | DQN | BootDQN | NoisyNet | BootDQN-IDS | BEBU | BEBU-UCB | BEBU-IDS | OEB3 |
| Scores | 0 | 100 | 3 | 0 | 0 | 0 | 0 | 0 |

Our method does not have a good performance on Montezuma's Revenge (see Table 7) because the epistemic uncertainty-based methods are not particularly tweaked for this domain. IDS, NoisyNet and BEBU-based methods all fail in this domain and score zero. Bootstrapped DQN achieves 100 points, which is also very low and does not indicate successful learning in Montezuma's revenge. In contrast, the bonus-based methods achieve significantly higher scores in this tasks (e.g., RND achieves 8152 points). However, according to Taiga et al. (2020) and Table 1, NoisyNet and IDS significantly outperform several strong bonus-based methods evaluated by the mean and median scores of 49 Atari games.

