# OpenReview forum: "Optimistic Exploration with Backward Bootstrapped Bonus for Deep Reinforcement Learning"
_ICLR.cc/2021/Conference — Reject_

### Official Review · AnonReviewer4 · 2020-10-24
**An interesting paper which applies optimism principle to practical algorithms**

**Rating:** 6
**Confidence:** 3

**Review:**

This paper studies optimistic exploration for deep reinforcement learning. They propose an algorithm called OEB3, which utilizes the disagreement of bootstrapping Q-functions as the confidence bonus to guide exploration. They show that their confidence bonus matches the confidence bonus of optimistic LSVI in linear setting. After that, they evaluate their algorithm on plenty of benchmark problems including Mnist maze and 49 Atari games. Their algorithm outperforms other SOTA exploration approaches.

Pros:

- The optimistic exploration problem is very important for the RL community. From the theoretical perspective, the classical $\epsilon$-greedy exploration method has been shown to be inefficient, and UCB-based algorithms are shown to be sample efficient with regret guarantee. However, how to apply these theoretical insights into practical algorithms is still an open problem. The paper proposes a method that uses bootstrapping to construct confidence bonus, and connects the theoretical findings to practical algorithms.

- The experimental results on 49 Atari games are rich enough to show the improvement of the algorithm, with visualization results and ablation studies.

Other comments:

- The connection with optimistic LSVI is interesting but not satisfying. Theorem 2 assumes that $\epsilon$ follows the standard Gaussian distribution. Why does this assumption hold? A more natural and reasonable assumption is that the rewards and transitions are sampled from a Gaussian distribution. Can theorem 2 still hold under such assumptions?


-------Post Rebuttal---------


Thanks for the feedback from the authors. After the rebuttal and discussion period, I believe that the contribution of the paper are mainly empirical. The Theorems (Theorem 1 and 2) are proposed to show the intuition of the bootstrapping and backward-update methods, and to connect the algorithm with recent theoretical results. During the discussion, there are mainly three concerns about the theorems among reviewers.

- Firstly, the algorithm uses eps-greedy to help exploration, which is theoretically unclear.
- Secondly, the theorems focus on episodic setting, while the algorithm is designed for discounted setting.
- Lastly, it is unclear whether the algorithm is a frequentest approach or Bayesian.

I believe that the former two concerns can be addressed in the following way, and don't weaken the contribution of the paper.
- From the theoretical perspective, the feedback from the authors does tackle this problem. The eps-greedy will only add an epsilon-term to the final regret. I am eager to see the discussion in the final version. From the experimental view, the authors also claim that all the algorithms in the experiments use eps-greedy to help exploration. In that case, the improved performance of their algorithms are mainly due to their bootstrapping and backward-update methods, instead of eps-greedy trick. Besides, I think applying several widely-used tricks (such as eps-greedy) to the algorithm is acceptable, which makes the results comparable to other methods that also use the tricks.
- To the best of my knowledge, there is currently no paper studying linear MDP in the discounted setting from the theoretical perspective. I think this is the reason why this paper studies the connection of the theorems in episodic setting. Meanwhile, it is hard to conduct experiments in episodic setting (In that case, the algorithm needs to learn the value function of all the possible horizon). Moreover, we can reduce an MDP with discounted rewards to episodic MDP by setting the effective horizon $H = \frac{1}{1-\gamma}$. This means that the theorem for episodic MDP can hold in the discounted setting with slight modification. Maybe such discussion should be added to the paper.

However, for the last concern, I am still puzzled whether the algorithm is a frequentest approach or Bayesian, as there are many unclear statements.

Overall, I believe the main contribution of the paper is from the empirical perspective. The theorem is intended to show the intuition of the algorithm and to connect the approach to the recent theoretical results. The experimental results look nice in general. I am a little disappointed that the authors missed the comparison with two related literatures in the initial version. I agree with R5 that these results may be done under a time-crunch. As a result, I change my score to 6, and I hope the above problems can be addressed in the next version.

---

> ### Author Response · Authors · 2020-11-23
> **Reply to AnonReviewer 4**
>
> We appreciate the valuable review and suggestions. We have revised our work accordingly. In what follows, we address the concern in detail.
>
> Note that our methods, Optimistic LSVI, and Bootstrapped DQN, are all frequentist approaches. The Bootstrapped DQN is connected to the Bayes approach with the uninformative prior (Friedman J. et al. 2001). In our analysis, we use Bayes setting as a simplification to motivate our algorithm. We highlight that our algorithm does not require such a setting. A similar motivation approach also arises in the recent work on the worst-case regret of randomized value function approaches (Russo, D. 2019).
>
> In addition, note that our analysis exploits the Gaussian likelihood of the target given the state, action, and parameter of the value function, which serves only for the motivation. Hence, as long as such a Gaussian likelihood holds, the result in Theorem 2 still holds. As an example, under the Gaussian transition and reward assumption, if we estimates the transition model instead of the value function, a similar result to Theorem 2 holds. On the other hand, the Gaussian assumption in transitions holds in the study of LQR. We refer to Kakade, S. et al. (2020) for a rigorous investigation for such a setting.
>
>
> **References**
>
> [Friedman J. et al. 2001] Jerome Friedman, Trevor Hastie and Robert Tibshirani. The elements of statistical learning. Springer series in statistics New York, 2001.
>
> [Russo, D. 2019] Russo, D. Worst-case regret bounds for exploration via randomized value functions. Advances in Neural Information Processing Systems. 2019.
>
> [Kakade, S. 2020] Kakade, S., Krishnamurthy, A., Lowrey, K., Ohnishi, M., Sun, W. Information Theoretic Regret Bounds for Online Nonlinear Control. Advances in neural information processing systems. 2020.

---

### Official Review · AnonReviewer1 · 2020-10-28
**Official Blind Review #3**

**Rating:** 6
**Confidence:** 4

**Review:**

This paper focuses on deep reinforcement learning and proposes a practical exploration algorithm called Optimistic Exploration algorithm with Backward Bootstrapped Bonus. Based on the Optimistic LSVI algorithm, the authors propose a new optimistic exploration bonus for general cases, similar to the optimistic exploration bonus in the LSVI algorithm under the linear MDP setting. Experiment results show that the QWR algorithm has better performance than previous algorithms.
This paper is well-written and easy to follow.  The main contributions are delivered:  Proposed optimistic exploration bonus for general cases; New deep reinforcement learning algorithm. However, I still have some concerns about this paper.
First, the actor-value function Q is the expected cumulative reward with discounted factor \gamma. However, it is usually set 0<\gamma<1 for infinite-horizon MDP and set \gamma=1 for episodic MDP or finite-horizon MDP. Besides, the LSVI algorithm also set \gamma=1, and it seems strange that there is a discounted factor \gamma in the OEB3 algorithm.
Second, in the LSVI algorithm, the agent chooses action by the totally greedy policy with the state-actor value function. In the OEB3 algorithm, the agent chooses action by the \epsilon-greedy policy. Even the optimistic exploration bonus in the OEB3 algorithm is the same as the LSVI algorithm under the linear MDP setting, theoretic proof of LSVI may not support the OEB3 algorithm.
Finally, in the experiment, it seems that the action space is relatively small. It is better to do more experiments with a broader action set and show the OEB3 algorithm's performance in those experiments.

---

> ### Author Response · Authors · 2020-11-23
> **Reply to AnonReviewer 1, Part 2/2**
>
> **3. The action-space of experiments**
>
> >  Finally, in the experiment, it seems that the action space is relatively small. It is better to do more experiments with a broader action set and show the OEB3 algorithm's performance in those experiments.
>
> The action-space in Atari games is $18$, including noop, fire, up, right, left, down, up-right, up-left, down-right, down-left, up-fire, right-fire, left-fire, down-fire, up-right-fire, up-left-fire, down-right-fire, and down-left-fire. We evaluate all methods on 49 Atari games, and our method outperforms several strong baselines.  Although the action space may seem small compared with complicated real-life applications, such a size of the action space is standard as an evaluation environment. Other popular environments with discrete action spaces also have similar action spaces. For example, in DeepMind Lab, the actions include forward, back, strafe left/right, crouch, jump, and looking (up/down, left/right), and the actions in the MiniGrid maze include left, right, forward, pick up an object, drop the object, toggle (open doors, interact with objects), and done.
>
> In continuous control tasks, SUNRISE (Lee et al, 2020) is an extension of Bootstrapped DQN (Osband et al, 2016) and UCB exploration (Chen et al, 2017) to continuous action space with SAC algorithm. Empirically, SUNRISE outperforms several strong baselines (PETS, TD3, Dreamer, CURL, PlaNet, SLAC, DrQ, and RAD) in Mujoco and DeepMind Control Suite, which indicates bootstrapped posterior is also powerful in continuous action space. Moreover, our methods is also promising to be extended in continuous control by following the implementation of SUNRISE. We leave this extension in future research.
>
>
> **References**
>
> [Mnih et al, 2015] Mnih, V., Kavukcuoglu, K., Silver, D. et al. Human-level control through deep reinforcement learning. Nature, 518(7540), 529-533. 2015
>
> [Osband et al, 2016] Osband, I., Blundell, C., Pritzel, A., Van Roy, B. Deep exploration via bootstrapped DQN. Advances in neural information processing systems. 2016
>
> [Nikolov et al, 2019] Nikolov, N., Kirschner, J., Berkenkamp, F., Krause, A. Information-Directed Exploration for Deep Reinforcement Learning. International Conference on Learning Representations. 2019
>
> [Lillicrap et al, 2016] Lillicrap, T. P., Hunt, J. J., Pritzel, A. et al. Continuous control with deep reinforcement learning. International Conference on Learning Representations. 2016
>
> [Fujimoto et al, 2018] Fujimoto, S., Hoof, H., Meger, D. Addressing Function Approximation Error in Actor-Critic Methods. International Conference on Machine Learning. 2018
>
> [Schulman et al, 2017] Schulman, J., Wolski, F., Dhariwal, P., Radford, A., Klimov, O. Proximal policy optimization algorithms. arXiv preprint arXiv:1707.06347. 2017.
>
> [Chen et al, 2017] Chen, R. Y., Sidor, S., Abbeel, P., Schulman, J. UCB exploration via Q-ensembles. arXiv preprint arXiv:1706.01502. 2017
>
> [Lee et al, 2020] Lee, K., Laskin, M., Srinivas, A., Abbeel, P. SUNRISE: A simple unified framework for ensemble learning in deep reinforcement learning. arXiv preprint arXiv:2007.04938. 2020

---

> ### Author Response · Authors · 2020-11-23
> **Reply to AnonReviewer 1, Part 1/2**
>
> We appreciate the valuable review and suggestions. We have revised our work accordingly. In what follows, we address your concerns in detail.
>
> **1. The setting of $\gamma$**
>
> >  First, the actor-value function Q is the expected cumulative reward with discounted factor \gamma. However, it is usually set 0<\gamma<1 for infinite-horizon MDP and set \gamma=1 for episodic MDP or finite-horizon MDP. Besides, the LSVI algorithm also set \gamma=1, and it seems strange that there is a discounted factor \gamma in the OEB3 algorithm.
>
> We set $\gamma=0.99$ in our experiment for several empirical considerations. (1) Using $\gamma<1$ is a standard setup in the implementation of deep RL algorithms to ensure empirical performances, even for episodic MDPs. For example, the value-based methods such as DQN (Mnih et al., 2015, Pages 6), bootstrapped DQN (Osband et al., 2016, Appendix D.2), IDS (Nikolov et al., 2019, Appendix A) all set $\gamma=0.99$ in Atari games. The policy-gradient methods such as DDPG (Lillicrap et al., 2016, section 7), TD3 (Fujimoto et al., 2018, Table 3), and PPO (Schulman et al., 2017, Appendix A) also set $\gamma=0.99$ in Mujoco tasks. (2) Although Atari is an episodic task in our experiment, the episodic length is relatively long, typically with hundreds of steps. Using $\gamma<1$ ensures that the value functions are bounded, which improves the empirical performance of the algorithms.
>
> **2. $\epsilon$-greedy in OEB3**
>
> >  Second, in the LSVI algorithm, the agent chooses action by the totally greedy policy with the state-actor value function. In the OEB3 algorithm, the agent chooses action by the \epsilon-greedy policy. Even the optimistic exploration bonus in the OEB3 algorithm is the same as the LSVI algorithm under the linear MDP setting, theoretic proof of LSVI may not support the OEB3 algorithm.
>
> $\epsilon$-greedy in OEB3. We adopt $\epsilon$-greedy for the empirical performance of OEB3. In experiments, we observe that in the beginning of training, the bootstrapped Q-heads usually lack diversity. Hence, exploring based on bonus become prohibitive at the early stage of training. To this end, we use the $\epsilon$-greedy policy to enhance exploration in the early stage and diminish $\epsilon$ as the training evolves. We remark that such a trick is also utilized in several popular bootstrapped-DQN implementations, including the bootstrapped-DQN implementation at https://github.com/johannah/bootstrap_dqn, https://github.com/rrmenon10/Bootstrapped-DQN, and the official IDS (Nikolov et al, 2019) implementation at https://github.com/nikonikolov/rltf. Moreover, the baselines (i.e., BEBU, BEBU-UCB, BEBU-IDS) in our paper all use $\epsilon$-greedy in training with the same values of $\epsilon$ for a fair comparison.
>
> In terms of the theory, we highlight that the $\epsilon$-greedy in place of greedy will hinder the performance difference term $\langle \pi^k, Q^* - Q^{k} \rangle$, which is upper bounded by zero if $\pi^k$ is the greedy policy corresponding to $Q^k$. In contrast, if $\pi^k$ is the $\epsilon$-greedy policy, adding and subtracting the greedy policy yields an $\epsilon Q_{\max}$ upper bound, which propagates to an additional $O(\epsilon T)$ term in the regret. Therefore, if $\epsilon$ is sufficiently small, the algorithm attains the optimal $\sqrt{T}$-regret. We remark that in our experiments, the $\epsilon$-term diminishes to zero, which does not incur a large bias to the regret of greedy policy.

---

### Official Review · AnonReviewer3 · 2020-10-28
**An optimistic exploration method for DRL using backward bootstrapped bonus which outperforms previous exploration approaches in Mnist Maze and Atrari games.**

**Rating:** 7
**Confidence:** 4

**Review:**

The paper proposes a UCB-based optimistic exploration method for DRL, called Optimistic Exploration algorithm with Backward Bootstrapped Bonus (OEB3). The algorithm builds on the idea of optimistic exploration developed in the theoretical optimistic LSVI algorithm for linear MDP. In optimistic LSVI, the optimistic Q-value is backward updated for each episode with an UCB bonus at each step. OEB3 extends this backward update of UCB bonus to DRL by estimating the UCB bonus using bootstrapped Q-learning. OEB3 learns an ensemble of bootstrapped Q-values and estimates the UCB bonus as the variance of the Q-values. Another important part of OEB3 is to adapt episodic backward update to bootstrapped Q-learning. This is done by procedure called Bootstrapped Episodic Backward Update (BEBU) which extends previous episodic update idea to the ensemble of bootstrapped Q-values.

Experiments have be done in Mnist Maze and Atrari games comparing OEB3 with previous exploration methods and some OEB3 variants. Table 2 nicely summarizes the algorithm design difference of related algorithms and highlight the two important components of backward update and UCB bonus in OEB3, and the improvements clearly helps OEB3 to outperforms similar exploration approaches. However, it may be miss-leading to say that OEB3 outperforms all existing bonus-based methods since it actually performs worse than some prior methods in Montezuma’s Revenge.

I think this is a nice work given its connection to theory and the nice empirical performance, but I think the following questions about steps in the OEB3 algorithm should be answered.

- The paper claims to use bootstrapped Q-learning, but in the algorithm description, at each time only one episode E is sampled for all head. This is different from the original bootstrapped Q-learning design where different heads are learned using different samples with bootstrap. Are there some steps missing in the algorithm description or no bootstrapped samples are used? If all heads are learned from the same sample, does the diversity among different heads only depend on their initialization?

- In training, the action is selected by randomly picking a head, i.e. the Thompson sampling approach. However, in evaluation the action seems to be selected by majority vote. Is there a reason for this difference in training and evaluation? Does random sampling help exploration in training?

- In algorithm description, epsilon-greedy is actually used in OEB3 during training. The use of epsilon-greedy is different from other exploration methods which guide exploration by bonus or other randomness. Why does OEB3 need epsilon-greedy given its exploration bonus design? In experiments, do other baselines also use epsilon-greedy?

---

> ### Author Response · Authors · 2020-11-23
> **Reply to AnonReviewer 3, Part 2/2**
>
> **4. $\epsilon$-greedy in OEB3**
>
> > In algorithm description, epsilon-greedy is actually used in OEB3 during training. The use of epsilon-greedy is different from other exploration methods which guide exploration by bonus or other randomness. Why does OEB3 need epsilon-greedy given its exploration bonus design? In experiments, do other baselines also use epsilon-greedy?
>
> $\epsilon$-greedy in OEB3. We adopt $\epsilon$-greedy for the empirical performance of OEB3. In experiments, we observe that in the beginning of training, the bootstrapped Q-heads usually lack diversity. Hence, exploring based on bonus become prohibitive at the early stage of training. To this end, we use the $\epsilon$-greedy policy to enhance exploration in the early stage and diminish $\epsilon$ as the training evolves. We remark that such a trick is also utilized in several popular bootstrapped-DQN implementations, including the bootstrapped-DQN implementation at https://github.com/johannah/bootstrap_dqn, https://github.com/rrmenon10/Bootstrapped-DQN, and the official IDS (Nikolov et al, 2019) implementation at https://github.com/nikonikolov/rltf. Moreover, the baselines (i.e., BEBU, BEBU-UCB, BEBU-IDS) in our paper all use $\epsilon$-greedy in training with the same values of $\epsilon$ for a fair comparison.
>
> In terms of the theory, we highlight that the $\epsilon$-greedy in place of greedy will hinder the performance difference term $\langle \pi^k, Q^* - Q^{k} \rangle$, which is upper bounded by zero if $\pi^k$ is the greedy policy corresponding to $Q^k$. In contrast, if $\pi^k$ is the $\epsilon$-greedy policy, adding and subtracting the greedy policy yields an $\epsilon Q_{\max}$ upper bound, which propagates to an additional $O(\epsilon T)$ term in the regret. Therefore, if $\epsilon$ is sufficiently small, the algorithm attains the optimal $\sqrt{T}$-regret. We remark that in our experiments, the $\epsilon$-term diminishes to zero, which does not lead to a large bias to the regret of greedy policy.
>
> **References**
>
> [Osband et al, 2016] Osband, I., Blundell, C., Pritzel, A., Van Roy, B. Deep exploration via bootstrapped DQN. Advances in neural information processing systems. 2016
>
> [Chen et al, 2017] Chen, R. Y., Sidor, S., Abbeel, P., Schulman, J. UCB exploration via Q-ensembles. arXiv preprint arXiv:1706.01502. 2017
>
> [Osband et al, 2018] Osband, I., Aslanides, J., Cassirer, A. Randomized prior functions for deep reinforcement learning. In Advances in Neural Information Processing Systems. 2018
>
> [Nikolov et al, 2019] Nikolov, N., Kirschner, J., Berkenkamp, F., Krause, A. Information-Directed Exploration for Deep Reinforcement Learning. International Conference on Learning Representations. 2019

---

> ### Author Response · Authors · 2020-11-23
> **Reply to AnonReviewer 3, Part 1/2**
>
> We appreciate the valuable review and suggestions. We have revised our work accordingly. In what follows, we address the concerns in detail.
>
> **1. On the unclear statement**
>
> > However, it may be miss-leading to say that OEB3 outperforms all existing bonus-based methods since it actually performs worse than some prior methods in Montezuma’s Revenge.
>
> We agree with the reviewer that the statement "outperforms existing bonus-based methods" is inaccurate. We change it to "outperforms several strong bonus-based methods evaluated by the mean and median score of 49 Atari games." Here the bonus-based methods refers to the count-based and curiosity-driven approaches. We refer to Section 1 for a detailed comparison.
>
> **2. Different heads are learned using same samples**
>
> > The paper claims to use bootstrapped Q-learning, but in the algorithm description, at each time only one episode E is sampled for all head. This is different from the original bootstrapped Q-learning design where different heads are learned using different samples with bootstrap. Are there some steps missing in the algorithm description or no bootstrapped samples are used? If all heads are learned from the same sample, does the diversity among different heads only depend on their initialization?
>
> Despite the fact that Bootstrapped DQN is proposed to train each head with independent samples, we remark that in tackling Atari games, Osband et al. (2016) observes that the bootstrapping does not contribute much in performance. Empirically, Bootstrapped DQN uses the same samples to train all Q-heads in each training step. This empirical simplification is also adopted by Chen et al, (2017), Osband et al, (2018) and Nikolov et al, (2019), which are based on Bootstrapped DQN. In our experiments, we use such an implementation simplification for all the methods that we compare, i.e., BEBU, BEBU-UCB, BEBU-IDS, and OEB3. We revised our paper to highlight this point in section 5.1. We refer to Appendix D of Osband et al. (2016) for a detailed discussion, and quote a relevant discussion by Osband et al. (2016) in the sequel.
>
> "In this setting all network heads share all the data, so they are not actually a traditional bootstrap at all. ... We have several theories for why the networks maintain significant diversity even without data bootstrapping in this setting. ... First, they all train on dierent target networks. This means that even when facing the same $(s, a, r, s')$ datapoint this can still lead to drastically dierent $Q$-value updates. Second, Atari is a deterministic environment, any transition observation is the unique correct datapoint for this setting. Third, the networks are deep and initialized from dierent random values so they will likely find quite diverse generalization even when they agree on given data. Finally, since all variants of DQN take many many frames to update their policy, it is likely that even using $p = 0.5$ they would still populate their replay memory with identical datapoints. This means using $p = 1$ to save on minibatch passes seems like a reasonable compromise and it doesn’t seem to negatively aect performance too much in this setting."
>
> **3. Majority-vote evaluation**
>
> > In training, the action is selected by randomly picking a head, i.e. the Thompson sampling approach. However, in evaluation the action seems to be selected by majority vote. Is there a reason for this difference in training and evaluation? Does random sampling help exploration in training?
>
> We remark that the majority-vote approach for evaluation follows the same evaluation method as in Bootstrapped DQN. As pointed out in section 6.4 of Osband et al. (2016), bootstrapped DQN may be beneficial as a purely exploitative policy in evaluation. One can combine all the heads into a single ensemble policy, for example by choosing the action with the most votes across heads. We refer to Osband et al. (2016) for the detailed analysis of such an approach.  Our paper follows the same approach as Bootstrapped DQN and adopts the majority-vote evaluation for our baselines (BootDQN, BootDQN-IDS, BEBU, BEBU-UCB, BEBU-IDS) and OEB3 for fair comparison.
>
> Meanwhile, we do not use the majority-vote policy in training for several reasons. (a) A purely exploitative policy prevent the agents from effectively exploring the environment. In contrast, different Q-functions from different Q-heads allow the agent to explore the environment efficiently. (b) By following a single head in an episode, the agent can perform temporally-extended exploration. We revised our paper accordingly to add the details in section 5.1.

---

### Official Review · AnonReviewer2 · 2020-10-28
**Combining optimism with Deep RL**

**Rating:** 6
**Confidence:** 4

**Review:**

The authors of the submission "Optimistic Exploration with Backward Bootstrapped Bonus for Deep Reinforcement Learning" draw inspiration from the theoretical reinforcement learning literature to propose an optimism based bonus for deep q learning. The idea is to compute an optimistic bonus based on a Q function ensemble, and use this to augment the present reward signal and the future return estimator (future optimism) to yield an algorithmic approach that reduces to LSVI in the case of linear MDPs (or at least reduces to something akin to LSVI in that case).

The authors then introduce a generic procedure (BEBU) that can be plugged into a variety of Q learning algorithms to produce an "optimistic" bootstrapped backward episodic update. Crucially the order of the updates matters, since the uncertainty bonuses should be propagated backwards in time. This is a very nice poin: the uncertainty propagation should be done backwards.

The authors are right in my opinion to claim their work represents a welcome addition to the emerging literature that proposes the use of uncertainty bonuses around the value function as opposed to myopic uncertainty bonuses only at the immediate reward level. There exist other recent works that introduce similar ideas (bringing in bonuses for the future uncertainty as opposed to solely penalizing immediate  ) are some missing citations in the related work section, most notably "On optimism in model based reinforcement learning" (using value optimism in model based RL and deep RL), "Efficient model based RL through optimistic policy search and planning"(optimism and GPs in model based RL), and also SUNRISE which looks awfully related to BEBU-UCB.

The experimental results of this work are strong. It is nevertheless unclear how much of these results are the consequence of accessibility to massive computing resources. I would like to see the paper positioned more faithfully within the relevant optimism-at-value-level literature, even though these works are model based in nature. It would also be very useful to have algorithm boxes for the different methods or method templates that the authors describe in the text. It is hard to follow what they intended to say or at least a table listing succinctly in a reader friendly way what the differences are between the different instantiations of the approach (OEB3, BEBU, BEBU-UCB, etc ...).

---

> ### Author Response · Authors · 2020-11-23
> **Reply to AnonReviewer 2, Part 2/2**
>
> **References**
>
> [Nix and Weigend, 1994] Nix, David A., and Andreas S. Weigend. Estimating the mean and variance of the target probability distribution. In Proceedings of IEEE International Conference on Neural Networks, vol. 1, pp. 55-60. IEEE, 1994.
>
> [Kalweit and Boedecker, 2017] Kalweit, Gabriel, and Joschka Boedecker. Uncertainty-driven imagination for continuous deep reinforcement learning. Conference on Robot Learning. 2017.
>
> [Buckman et al. 2018] Buckman, Jacob, et al. Sample-efficient reinforcement learning with stochastic ensemble value expansion. Advances in Neural Information Processing Systems. 2018.
>
> [Sekar et al, 2020] Sekar, Ramanan, et al. Planning to Explore via Self-Supervised World Models. International Conference on Machine Learning. 2020
>
> [Hafner et al, 2020] Hafner, Danijar, et al. Dream to Control: Learning Behaviors by Latent Imagination. International Conference on Learning Representations. 2020.
>
> [Ball et al, 2020] Ball, P., Parker-Holder, J., Pacchiano, A., Choromanski, K. and Roberts, S., 2020. Ready Policy One: World Building Through Active Learning. arXiv preprint arXiv:2002.02693.
>
> [Pacchiano et al, 2020] Pacchiano, A., Ball, P., Parker-Holder, J., Choromanski, K., Roberts, S. On optimism in model-based reinforcement learning. arXiv:2006.11911. 2020
>
> [Agrawal and Jia, 2017] Agrawal, Shipra, and Randy Jia. Posterior sampling for reinforcement learning: worst-case regret bounds. Advances in Neural Information Processing Systems. 2017.
>
> [Curi et al, 2020] Curi, Sebastian, Felix Berkenkamp, and Andreas Krause. Efficient model-based reinforcement learning through optimistic policy search and planning. Advances in Neural Information Processing Systems. 2020.
>
> [Chen et al, 2017] Chen, R. Y., Sidor, S., Abbeel, P., Schulman, J. UCB exploration via Q-ensembles. arXiv preprint arXiv:1706.01502. 2017
>
> [Lee et al, 2020] Lee, K., Laskin, M., Srinivas, A., Abbeel, P. SUNRISE: A simple unified framework for ensemble learning in deep reinforcement learning. arXiv preprint arXiv:2007.04938. 2020

---

> ### Author Response · Authors · 2020-11-23
> **Reply to AnonReviewer 2, Part 1/2**
>
> We appreciate the valuable review and suggestions. In what follows, we address the concerns in detail.
>
> **1. Related model-based methods**
>
> > There exist other recent works that introduce similar ideas (bringing in bonuses for the future uncertainty as opposed to solely penalizing immediate ) are some missing citations in the related work section, most notably "On optimism in model based reinforcement learning" (using value optimism in model based RL and deep RL), "Efficient model based RL through optimistic policy search and planning"(optimism and GPs in model based RL), and also SUNRISE which looks awfully related to BEBU-UCB.
>
> We thank the reviewer's suggestion on the related model-based methods. We highlight that our work is model-free, which is relevant though parallel to the model-based optimistic approaches. The main difference is that our work estimates an optimistic Q-function to guide exploration. In contrast, the model-based approach typically estimates a confidence set of transitions, and chooses the transition optimistically from the confidence set at each iteration for planning and exploration. We revise our Related Works (Section 4) to include a discussion on model-based approaches with optimism.
>
> "Beyond model-free methods, model-based RL also uses optimism for planning and exploration (Nix and Weigend, 1994). Model-assisted RL (Kalweit and Boedecker, 2017) uses ensembles to make use of artificial data only in cases of high uncertainty. Buckman et al. (2018) uses ensemble dynamics and Q-functions to use model rollouts when they do not cause large errors. Planning to explore (Sekar et al., 2020) seeks out future uncertainty by integrating uncertainty to Dreamer (Hafner et al., 2020). Ready Policy One (Ball et al., 2020) optimizes policies for both reward and model uncertainty reduction. Noise-Augmented RL (Pacchiano et al., 2020) uses statistical bootstrap to generalize the optimistic posterior sampling (Agrawal and Jia, 2017) to DRL. Hallucinated UCRL (Curi et al., 2020) reduces optimistic exploration to exploitation by enlarging the control space. The model-based RL needs to estimate the posterior of dynamics, while OEB3 relies on the posterior of Q-functions."
>
> SUNRISE (Lee et al., 2020) nicely extends Bootstrapped DQN (Osband et al, 2016) and UCB exploration (Chen et al. 2017) to continuous control through confidence reward and weighted Bellman backup. We have compared with Chen et al. (2017) with backward update in experiments. Moreover, OEB3 is also promising to be extended in continuous control by following the implementation of SUNRISE. We leave this extension in future research. We have added this reference in the revised paper.
>
> **2. On the massive computing resources**
>
> > The experimental results of this work are strong. It is nevertheless unclear how much of these results are the consequence of accessibility to massive computing resources.
>
> We would like to clarify that as far as we are concerned, our performance comparison seems to be fair in terms of the computational resources for the following reasons. (1) In the experiments, we do not use distributed architecture for OEB3 and other baselines. We conduct experiments on all the exploration methods on the same machine with one RTX-2080Ti GPU per task. (2) We compare the performance of OEB3, BEBU, BEBU-UCB, and BEBU-IDS by training them with the same number of training frames (20M frames), which guarantees a fair comparison in terms of exploration.
>
> **3. On the algorithm boxes for the different methods**
>
> >  It would also be very useful to have algorithm boxes for the different methods or method templates that the authors describe in the text. It is hard to follow what they intended to say or at least a table listing succinctly in a reader friendly way what the differences are between the different instantiations of the approach (OEB3, BEBU, BEBU-UCB, etc ...).
>
> We thank the reviewer's suggestions on the algorithm boxes for the different methods. We revised the paper to include additional algorithmic descriptions for three baselines (i.e., BEBU, BEBU-UCB, and BEBU-IDS). We use blue color to highlight the difference between baselines. Please refer to Algorithm 3 of Appendix B in the revised manuscript for the details.

---

### Official Review · AnonReviewer5 · 2020-11-07
**Incremental paper. Needs a clearer presentation and a more thorough empirical analysis.**

**Rating:** 4
**Confidence:** 4

**Review:**

#### SUMMARY

The paper proposes a strategy for computing and propagating uncertainty to improve exploration in deep reinforcement learning. While the architecture necessary for computing the uncertainty is reliant on the framework of an existing deep RL algorithm — bootstrap DQN — the idea for propagation of this uncertainty to yield effective behaviour, or optimistic value estimates, hinges upon strategies used in the finite-horizon algorithms — constraining learning to be at the timescale of episodes (instead of samples).

The proposed main algorithm OEB3 — Optimistic Exploration with Backward Bootstrapped Bonus — combines ideas from existing literature to promote optimistic value estimates, and hence, as a product, effective exploratory behaviour. Particularly, I think it brings together 4 ideas from 4 different referenced papers:
1. a learning architecture that implements non-parametric Bayesian Value Iteration (as done in Osband et. al. 2016)
2. uncertainty estimation (as done in Chen et. al. 2019)
3. uncertainty propagation by constraining learning timescale (as done in Lee et. al. 2019 and Jin et. al. 2019)
4. uncertainty propagation by bootstrapping from optimistic estimates (as done in Jin et. al. 2019)

The paper also highlights that due to the uncertainty propagation achieved by incorporating ideas from (3) and (4) the induced exploratory behaviour is more effective, and hence the sample complexity of learning is reduced. Therefore, instead of the more common 200M frames used for training deep RL agents, 20M frames are used for training in the experiments here.


#### STRENGTHS

I think the paper is a fine example of research that builds on existing ideas in the field. The relevant work is discussed and the paper gradually builds up to the core proposal. The empirical analysis is comprehensive across the Atari suite, in order to be wary of recent work that suggests design of deep RL exploration methods may be overfitting to performance measures in a skewed subset of Atari (although, as it seems to be de-facto standard of the field, it averages across 5 seeds). Additionally, as the algorithm does have many components, the paper does a good job of explaining the strategy employed for uncertainty propagation well in the text.


#### WEAKNESSES

While I do think incremental research that builds on existing work is very valuable, I think the paper can be improved due to the following three key limitations:
1.  the presentation style currently obfuscates the incremental nature of the paper. For instance
	(a) the uncertainty estimation strategy is as proposed by Chen et. al. 2019, but the presentation seems to suggest it is novel. While Chen et. al. 2019 do not propose to “propagate” it during bootstrap as well, the bonus computed is the straightforward empirical standard deviation, as proposed by them.
	(b) the informal Theorem 1 (and corresponding formal Theorem 2) I think are well known in that the bonus used by UCB algorithms for linear regression is proportional to the posterior variance of bayesian linear regression. I am having a hard time seeing this as an insightful contribution. Further, LSVI-UCB is a frequentist solution approach and bootstrap DQN a non-parametric Bayesian approach — combining the two definitely warrants some discussion.
	(c) BEBU is essentially EBU as proposed in Lee et. al. 2019 with bonuses added. I do not think the text presents it so. Further, “faithfully follows the backward update of optimistic LSVI” may be a stretch as the optimistic LSVI bootstrap estimates are post learning at every step of backward induction.
2. While I understand a complete empirical comparison to the many existing deep RL methods can be very expensive, the literature review does miss methods proposed with a similar ethos — propagation of uncertainty. I think they need to be discussed and compared against
	(a) Bayesian Deep Q-Networks : Azizzadenesheli et. al. 2019 [https://arxiv.org/pdf/1802.04412.pdf]
	(b) Uncertainty Bellman Equation: Osband et. al. 2018 [https://arxiv.org/pdf/1709.05380.pdf]
3. Some of the proposals made are unclear and left unexplored/discussed.
	(a) bonus in bootstrap target - different from LSVI-UCB
	(b) $\epsilon$-greedy with bootstrap dqn - non-parametric “optimistic” bayesian approach with dithering?
	(c) computation of $\tilde{B}^k$ (I may have missed this, but presumably its the empirical standard deviation with the target networks of the ensembles)

Further, I think some discussion in the main/appendix are warranted in terms of the empirical setup:
1. the buffer is a circular buffer of size 1M — is it 1M trajectories or 1M samples?
2. handling of episode cutoffs (which I presume are used), for bootstrapping at the end of the trajectory.
3. the magnitude of $\alpha_1$ and $\alpha_2$ are really small — the scale of the bonuses would be interesting to look into as well.


#### QUESTIONS
1. I understand the bonus proposed is an empirical surrogate to promote optimistic values (with respect to the seen data) — but ideally, do we not need optimistic values that include q*. What is the guarantee that q^* is included in the set (as it seems to be for the example in Figure 1).
2. What do you see are the key advantages of the proposed approach? Is scalability by constraining learning to be at the episode level an issue in practical applications? (I think this also can be discussed in the paper).


#### SUGGESTIONS
My concrete suggestions to improve the paper while keeping the core idea the same is two fold:
1. a more thorough empirical section comparing to methods designed for uncertainty propagation
2. a rewrite which reflects that the components of the core idea are existing proposals in literature in the case of both bonus estimation and the EBU algorithm.
3. I think it is surprising that OEB3 is poorer that Bootstrap DQN in Montezuma’s revenge - presumably an increased number of training steps should alleviate the discrepancy; if not, the bonuses propagated by OEB3 may actually hinder performance which I think warrants acknowledgment/discussion.

#### Minor typos
1. Mnist  —> MNIST
2. Medium —> Median (Table 1)
3. Presumably Figures 2 and 3 are bonuses during learning — but a phrase in Section 5.2 says “trained OEB3”.
4. Section 5.2: incentive —> incentivize



## POST-REBUTTAL

I really appreciate the author's engagement and response during the discussion phase to help me understand the paper better, and the revisions incorporated in the paper.
But after much thought, I do not think the current form of the paper meets the bar for publication.

Here are my main concerns that I hope is useful for the next version or final submission.

I think the core idea of the algorithm is uncertainty propagation is necessary for inducing effective exploratory behaviour. This core idea is theoretically motivated from sound strategies for exploration in finite-horizon RL, but as the paper addresses the discounted problem setting with deep neural networks the soundness is traded-in for computational tractability — which is a fine choice.

But, currently the choices are presented in a confusing way, and the actual contributions of the algorithm are unclear: most importantly, is it a Frequentist approach to exploration or Bayesian?

- under the Frequentist approach setting, utilizing e-greedy seems justifiable just based on the reasoning of this is “the widely accepted practice in the field”.
- under the Bayesian viewpoint, which is the crux of the architecture used here (Bootstrap DQN), the attempted theoretical connections in the paper (Theorem 1) and the practical algorithm proposed (based on the ideas of Optimistic LSVI), do not provide a clear picture.

To show theoretical soundness the algorithm is anchored to a Bayesian architecture and theoretical uncertainty connection, but for practical performance purposes the paper leverages reasoning from Frequentist Deep RL methods. Maybe this is a step in the right direction, and the extensive empirical results do seem to suggest it is effective, but the presentation is unclear. From the current draft:
- “OEB3 relies on the posterior of Q-functions” — Bayesian
- “UBE uses posterior sampling for exploration, whereas OEB3 uses optimism for exploration” — Frequentist

Therefore, this can be improved and presented more clearly to communicate the idea.

---

> ### Author Response · Authors · 2020-11-23
> **Reply to AnonReviewer 5, Part 5/5**
>
> **8. On a clarification to the figures.**
>
> > Presumably Figures 2 and 3 are bonuses during learning — but a phrase in Section 5.2 says “trained OEB3”.
>
> In figure 2, we use a trained OEB3 agent to interact with the environment for an episode and record the UCB-bonuses at each step. In Figure 3, we record the mean of bonuses along the training process. We revised our work and added detailed descriptions to the figures.
>
> **References**
>
> [Friedman J. et al. 2001] Jerome Friedman, Trevor Hastie and Robert Tibshirani. The elements of statistical learning. Springer series in statistics New York, 2001.
>
>
> [Chen et al. 2017] Richard Y Chen, Szymon Sidor, Pieter Abbeel, and John Schulman. UCB Exploration via Q-Ensembles. arXiv preprint arXiv:1706.01502, 2017
>
> [Lee et al. 2019] Su Young Lee, Choi Sungik, and Sae-Young Chung. Sample-Efficient Deep Reinforcement Learning via Episodic Backward Update. Advances in Neural Information Processing Systems, 2019
>
> [O'Donoghue et al. 2018] Brendan O'Donoghue, Ian Osband, Remi Munos and V. Mnih. The Uncertainty Bellman Equation and Exploration. International Conference of Machine Learning. 2018
>
> [Azizzadenesheli et al. 2018] Kamyar Azizzadenesheli, Emma Brunskill, and Anima Anandkumar. Efficient Exploration through Bayesian Deep Q-Networks. Information Theory and Applications Workshop (ITA). 2018
>
> [Osband et al. 2016] Ian Osband, Charles Blundell, Alexander Pritzel, Benjamin Van Roy. Deep Exploration via Bootstrapped DQN. Advances in Neural Information Processing Systems, 2016
>
> [Osband et al. 2019] Ian Osband, Benjamin Van Roy, Daniel J. Russo, Zheng Wen. Deep Exploration via Randomized Value Functions. Journal of Machine Learning Research, 2019.
>
> [Nikolov et al, 2019] Nikolov, N., Kirschner, J., Berkenkamp, F., Krause, A. Information-Directed Exploration for Deep Reinforcement Learning. International Conference on Learning Representations. 2019
>
> [Jin et al. 2020] Chi Jin, Zhuoran Yang, Zhaoran Wang, and Michael I Jordan. Provably Efficient Reinforcement Learning with Linear Function Approximation. In Proceedings of Thirty Third Conference on Learning Theory. 2020
>
> [Kiran et al. 2020] Kiran, B. R., Sobh, I., Talpaert, V., Mannion, P., Sallab, A. A. A., Yogamani, S., Pérez, P. Deep reinforcement learning for autonomous driving: A survey. arXiv preprint arXiv:2002.00444. 2020

---

> ### Author Response · Authors · 2020-11-23
> **Reply to AnonReviewer 5, Part 4/5**
>
> **4. On clarifications to the empirical setup.**
>
> > Is it 1M trajectories or 1M samples
>
> (1) The buffer contains 1M transitions, which is also the default setting in DQN, Bootstrapped DQN, EBU, and IDS. In OEB3, each transitions are stored as $(s,a,r,s',done)$. where "done" indicates if this transition is the end of an episode. We use this flag to identify the end of an episode in training. We refer to Table 4 in Appendix D for the details.
>
> > Handling of episode cutoffs at the end of the trajectory
>
> (2) Handling of episode cutoffs. We set the value function at the end of the episode to be zero for backward propagation in our update. That said, at the end of an episode, the target consists of only the reward and the bonus. We refer to Section 3.2 of our manuscript for a detailed example.
>
> > The magnitude of $\alpha_1$ and $\alpha_2$ are really small
>
> (3) The magnitude of $\alpha_1$ and $\alpha_2$. We perform a hyper-parameter search of $\alpha_1$ and $\alpha_2$ on five popular tasks, including Breakout, Freeway, Qbert, Seaquest, and SpaceInvaders. Our current setting is the best choice on average. The value is relatively small to suit the backward update in training empirically. If the parameters $\alpha_1$ and $\alpha_2$ are large, then the large bonus accumulates along with the training process, eventually resulting in a large Q-value at the beginning of the episode and affecting the convergence of Q-learning. Meanwhile, we remark that the episodes are relatively long in Atari (which typically consist of hundreds of steps). Therefore, a small weight typically yields better performance empirically.
>
> **5. On the coverage of value function.**
>
> > The bonus proposed is an empirical surrogate to promote optimistic values — but ideally, do we not need optimistic values that include q*. What is the guarantee that q^* is included in the set (as it seems to be for the example in Figure 1)
>
> We highlight that Optimistic LSVI intrinsically captures the model uncertainty, which covers the true transition $P^*$. Such coverage is implicitly reflected by the value functions (as we do not estimate the transition model directly). Since the model uncertainty covers $P^*$, the value functions obtained after planning in our algorithm covers $Q^*$. In addition, the nonnegative bonus guarantees that the estimated $Q$-function is larger than $Q^*$. Meanwhile, the UCB bonus shrinks to zero as we observe more samples. We verify such a fact empirically in our experiments, suggesting that the estimated $Q$-function eventually converges to the optimum. Specifically, In Figure 3, we record the the mean of UCB-bonus of the training batch in the learning process. As more experiences of states and actions are gathered, the mean UCB-bonus reduces gradually, which indicates that the bootstrapped value functions concentrate around the optimal value and the epistemic uncertainty decreases. The optimistic Q-value gradually converges to optimal Q-value in exploration.
>
> **6. On the key advantage of the proposed approach and its scalability.**
>
> > What do you see are the key advantages of the proposed approach? Is scalability by constraining learning to be at the episode level an issue in practical applications?
>
> The key advantage of OEB3 is the optimism for exploration, which improves the sample efficiency of Q-learning. More importantly, motivated by the theoretic findings, we propose to propagate the bonus through time by backward update, which allows for deep exploration in MDPs, and leads to better empirical performances than the baselines. When handling non-episodic problems, a simple and efficient remedy is to cut the long sequence into several episodic experiences (with a proper handling of the cut-offs). This technique is often used in real-world applications with non-episodic setting, such as self-driving cars. As discussed in a recent survey (Kiran et al. 2020), using DRL method in self-driving needs to manually define the episode length with a finite horizon.
>
> **7. Performance on Montezuma's revenge.**
>
> > It is surprising that OEB3 is poorer that Bootstrap DQN in Montezuma’s revenge
>
> The score of 100 is reported in Osband et al. (2016), which we believe is prone to implementation. We use another popular bootstrapped-DQN implementation at https://github.com/nikonikolov/rltf and trains the policy for 200M frames, which scores zero on Montezuma's revenge.  In our released code at https://github.com/review-anon/OEB3 (run without --BEBU and --reward-type to train a bootstrapped-DQN agent), the final result of Bootstrapped DQN is also zero. Also, the score of 100 is relatively low, which does not indicate successful learning in Montezuma's revenge (scoring 100 indicates that the player does not pass the first room). In contrast, the bonus-based methods achieve significantly higher scores in Montezuma's revenge. As a result, both OEB3 and Bootstrapped DQN performs poorly in such a task. We refer to Appendix H for the failure analysis.

---

> ### Author Response · Authors · 2020-11-23
> **Reply to AnonReviewer 5, Part 3/5**
>
>
> (2) Bayesian-DQN (Azizzadenesheli et al. 2018) modifies the linear regression of the last layer in Q-network and uses Bayesian Linear Regression (BLR) instead, which estimates an approximated posterior of the Q-function. When interacting with the environment, Bayesian-DQN uses Thompson Sampling on the posterior to capture the uncertainty of Q-estimates. Our method is different from Bayesian-DQN in several aspects. (a) Bayesian-DQN does not propagate the uncertainty, which cannot perform deep exploration for MDPs. (b) OEB3 uses a bootstrapped Q-learning to calculate a bootstrapped distribution of Q-function, while Bayesian-DQN uses a parametric Gaussian BLR to update the posterior. (c) OEB3 is more computationally efficient because the Bayesian-DQN needs to calculate the inverse of a high dimensional matrix (512*512 in Atari) when updating the posterior.
>
> Empirically, since Bayesian-DQN is not evaluated in the whole Atari suit, we adopt the official release code in https://github.com/kazizzad/BDQN-MxNet-Gluon and make two modification for a fair comparison. First, we add the 30 no-op evaluation mechanism, which we use to evaluate OEB3 and other baselines in our work. Second, we set the frame-skip to 4 to be consistent with our baselines. We remark that inconsistency in the comparison might still exist since the original implementation of Bayesian-DQN is based on MX-Net Library, while OEB3 and other baselines are implemented with Tensorflow. We release the modified code in https://github.com/review-anon/Bayesian-DQN. We completed one-seed's training in 49 Atari games for 20M training frames by our submission of this rebuttal. The mean and median of evaluated scores are 216% and 24%, while OEB3 obtains mean and median at 765% and 50%. We will complete the experiments of Bayesian-DQN for more seeds and add the results of the comparison in the future revisions.
>
> **3. On the unclear proposals.**
>
> > Bonus in bootstrap target - different from LSVI-UCB
>
> (1) Bonus in the target. We highlight that OEB3 is equivalent with Optimistic LSVI with Bootstrapped DQN, and the difference in implementation comes from adapting to Bootstrapped DQN. Note that instead of maintaining an optimistic estimation of Q-function as in Optimistic LSVI, our work maintains the target-Q function by a Q-ensemble as a consequence of adopting Bootstrapped DQN. Hence, to recover the optimistic Q-function estimation for the backward update, we need to add the bonus for target Q-networks onto each head of the ensembles as the targets.
>
> > $\epsilon$-greedy with bootstrap dqn - non-parametric “optimistic” bayesian approach with dithering
>
> (2) $\epsilon$-greedy in OEB3. We adopt $\epsilon$-greedy for the empirical performance of OEB3. In experiments, we observe that in the beginning of training, the bootstrapped Q-heads usually lack diversity. Hence, exploring based on bonus become prohibitive at the early stage of training. To this end, we use the $\epsilon$-greedy policy to enhance exploration in the early stage and diminish $\epsilon$ as the training evolves. We remark that such a trick is also utilized in several popular Bootstrapped DQN implementations, including the Bootstrapped DQN implementation at https://github.com/johannah/bootstrap_dqn, https://github.com/rrmenon10/Bootstrapped-DQN, and the official IDS (Nikolov et al, 2019) implementation at https://github.com/nikonikolov/rltf. Moreover, the baselines (i.e., BEBU, BEBU-UCB, BEBU-IDS) in our paper all use $\epsilon$-greedy in training with the same values of $\epsilon$ for a fair comparison.
>
> In terms of the theory, we highlight that the $\epsilon$-greedy in place of greedy will hinder the performance difference term $\langle \pi^k, Q^* - Q^{k} \rangle$, which is upper bounded by zero if $\pi^k$ is the greedy policy corresponding to $Q^k$. In contrast, if $\pi^k$ is the $\epsilon$-greedy policy, adding and subtracting the greedy policy yields an $\epsilon Q_{\max}$ upper bound, which propagates to an additional $O(\epsilon T)$ term in the regret. Therefore, if $\epsilon$ is sufficiently small, the algorithm attains the optimal $\sqrt{T}$-regret. We remark that in our experiments, the $\epsilon$-term diminishes to zero, which does not incur a large bias to the regret of greedy policy.
>
> > Computation of $\tilde{\mathcal{B}}^k$. (I may have missed this, but presumably its the empirical standard deviation with the target networks of the ensembles)
>
> (3) Computation of $\tilde{\mathcal{B}}^k$. We compute the target bonus $\tilde{\mathcal{B}}^k$ by the empirical standard deviation of the target network ensembles. We refer to Section 3.2 for a revised explanation for the computation.

---

> ### Author Response · Authors · 2020-11-23
> **Reply to AnonReviewer 5, Part 2/5**
>
> > BEBU is essentially EBU as proposed in Lee et. al. (2019) with bonuses added. I do not think the text presents it so.
>
> (3) We remark that BEBU is the combination of EBU and Bootstrapped DQN. Meanwhile, our proposed OEB3 is BEBU with the bonus propagated in the backward update. We remark that although the backward update is scarcely studied in empirical work (EBU is the only work as far as we are concerned), such an approach is a standard approach in the theoretical analysis of sample efficient reinforcement learning and deserves more empirical investigation. Also, we remark that EBU is essentially Optimistic LSVI without the bonus added. The main difference between Optimistic LSVI and OEB3 is that the bonus in OEB3 is estimated by Bootstrapped DQN to suit the neural network parameterization.
>
> In addition, we highlight that our contribution also includes the extension of backward update from ordinary $Q$-learning to bootstrapped DQN empirically. The implementation of BEBU in our work is different from EBU and Bootstrapped DQN in several aspects.  (a) BEBU requires extra tensors to store the UCB-bonus for immediate reward and next-$Q$ value, which are integrated into training to propagate uncertainties. (b) Integrating uncertainty into BEBU needs special design in training. Since $Q^k$ in $t$-th step is updated optimistically at the state-action pair $(s_{t+1}, a_{t+1})$ during the backward update, we introduce the mask $1_{a'\neq a_{t+1}}$ to ignore the bonus of next-$Q$ value in the update of $Q^k$ when $a'$ is equal to $a_{t+1}$. We revised the manuscript to add this discussion in Section 3.2.
>
> > Further, “faithfully follows the backward update of optimistic LSVI” may be a stretch as the optimistic LSVI bootstrap estimates are post learning at every step of backward induction.
>
> (4) On the difference in implementation details between Optimistic LSVI and OEB3. We highlight that the focus of Optimistic LSVI is theoretical. In addition, the theory in Optimistic LSVI requires a strong linear assumption in the transition dynamics. To make it works empirically, we make several adjustments in the implementation details.
> - In each training step of optimistic-LSVI, all historical samples are utilized to update the weight of Q-function and calculate the confidence bonus. While in OEB3, we use samples from a batch of episodic trajectories from the replay buffer in each training step. Such a difference in implementation is imposed to achieve computational efficiency.
> - In OEB3, the target-network has a relatively low update frequency, whereas, in Optimistic LSVI, the target Q function is updated in each iteration. We highlight that such implementation techniques are commonly used in most existing (off-policy) Deep RL algorithms, including DQN, BootDQN, DDPG, SAC. We revised the manuscript accordingly and highlight such differences between optimistic LSVI and OEB3 at the end of section 3.2.
>
> **2. On the discussion and empirical comparison on two related works (O'Donoghue et al. 2019 and Azizzadenesheli et al. 2018).**
>
> > I think they need to be discussed and compared against (a) Bayesian Deep Q-Networks (b) Uncertainty Bellman Equation
>
> We thank the reviewer for the references, which are highly related to our work. We include comprehensive comparisons in our revised manuscript
> and briefly discuss these works in the sequel. We refer to the table for a summary of the comparisons.
>
> (1) Uncertainty Bellman Equation and Exploration (UBE) (O'Donoghue et al. 2018). UBE proposes an upper bound on the variance of the posterior distribution of the Q-values, which are further utilized for posterior sampling in exploration. Specifically, when interacting with the environment, the agent chooses actions according to a posterior of Q-values, where the variance is the estimated upper bound. Our method is different from UBE in several aspects. (a) UBE uses posterior sampling for exploration, whereas OEB3 uses optimism for exploration. (b) OEB3 utilizes the episodic backward update for training, while UBE uses forward update with a seperate propagation of uncertainty. UBE does not propagate the uncertainty by the backward update.  As a result, our approach performs better empirically. (c) UBE needs to calculate the inverse of a high dimensional matrix (512*512 in Atari) for each update, which is computationally inefficient.
>
> Empirically, the UBE is evaluated by the whole Atari suit in the original paper. Since there is no released code from O'Donoghue et al. (2018), we directly adopt the scores reported in this paper. The mean and median evaluated score with 200M frames in Atari suit is 439.88% and 126.41%, respectively. Comparatively, the three baselines used in our paper (i.e., BootDQN, NoisyNet and BootDQN-IDS) achieves (mean, median) scores as (553%, 128%), (651%, 172%), and (757%, 187%), respectively. The three baselines in our paper outperform UBE. We refer to Table 1 in the revised manuscript for the comparisons.

---

> > ### Comment · AnonReviewer5 · 2020-11-24
> > **Response Part 2/5**
> >
> > 1.(3) I apologize for confusing BEBU and OBE3. But given my clarified understanding I think Section 3’s presentation can be improved to be clearer in the order of: (3.1) BEBU, (3.2) Bonus in OEB3, (3.3) OEB3.
> > Currently the introduction of how bonus is incorporated into OEB3 under Section 3.2 whose heading reads BEBU is a tad confusing.
> >
> > 2. I think the computational complexity of an inversion for a single head network may need to be contrasted with training n heads in a backward update fashion. In such a scenario, I am unsure which framework is more computationally tractable, but nonetheless, as mentioned before, OEB3’s empirical success is interesting and may make a valuable case for the idea of backwards update for uncertainty propagation. Further I think this sentence in your response is key and could be beneficial if highlighted in the paper — “UBE uses posterior sampling for exploration, whereas OEB3 uses optimism for exploration” — that although OEB3 is inspired based on the architecture of a a non-parametric Bayesian algorithm Bootstrap-DQN, it is an frequentist optimistic approach to exploration. In the same vein, incorporating the table at the beginning of Part1/5 of your response may be valuable.

---

> > > ### Author Response · Authors · 2020-11-25
> > > **Follow-up response to AnonReviewer5, Part 2/5**
> > >
> > > 1.(3) We want to introduce the UCB-Bonus in OEB3 first, and then present the method to propagate uncertainty through time by the backward update, which exploits the theoretical benefit of relevant analysis established by Jin et al. (2020). We change the heading of section 3.2 to "Uncertainty Backward in OEB3".
> > >
> > > 2.We agree with the reviewer's concern that the direct comparison of computational complexity is unnecessary because these two methods use different network architectures. We contrast the computation complexity as an observation of the experiments that we conduct and the empirical detail in the compared work (which uses the 512-dimensional output). We agree that such an issue depends on the architecture adopted for experiments. Nevertheless, we highlight that such an issue with the output dimension does not arise in our work. In addition, we thank the reviewer's suggestion on highlighting the difference. We have revised accordingly in our revision (Section 4). We also add the table in our response in the revised version (Appendix B).

---

> ### Author Response · Authors · 2020-11-23
> **Reply to AnonReviewer 5, Part 1/5**
>
> We appreciate the valuable suggestions and typos spotted by the reviewer. We have revised our work accordingly. We refer to table for a summary of comparisons in our rebuttal. In what follows, we address the concerns in detail.
>
> Methods | Bonus or Posterior Variance | Update Method | Uncertainty Characterization
> ---|:--:|:--:|:--:
> EBU (Lee et al. 2019)                      | -               | Backward update      | -
> Bootstrapped DQN (Osband et al. 2016)      | Bootstrapped    | On-trajectory update | Bootstrapped distribution
> BEBU (base of our work)                    | Bootstrapped    | Backward update      | Bootstrapped distribution
> Optimistic LSVI (Jin et al. 2020)          | Closed form     | Backward update      | Optimism
> OEB3 (our work)                            | Bootstrapped    | Backward update      | Optimism
> UBE (O'Donoghue et al. 2018)               | Closed form     | On-trajectory update | Posterior sampling
> Bayesian-DQN (Azizzadenesheli et al. 2018) | Closed form     | On-trajectory update | Posterior sampling
>
> **1. On the presentation of this paper.**
>
> >  The uncertainty estimation strategy is as proposed by Chen et. al. 2019, but the presentation seems to suggest it is novel. While Chen et. al. 2019 do not propose to propagate it during bootstrap, the bonus computed is the straightforward empirical standard deviation.
>
> (1) The relations between OEB3 and Chen et al. (2017). Chen et al. (2017) proposes to use the standard-deviation of bootstrapped Q-functions to measure the uncertainty. While the uncertainty measurement proposed by Chen et al. (2017) is similar to that of OEB3, our proposed OEB3 is different from Chen et al. (2017) in the following aspects.
> - The method proposed by Chen et al. (2017) does not propagate the uncertainty through time. In contrast, our approach propagates the uncertainty through time by the backward update, which allows for deep exploration for the MDPs.
> - The method proposed by Chen et al. (2017) does not use the bonus in the update of Q-functions. The bonus is computed when taking the actions. In contrast, we compute the UCB-bonus for the update of Q-function instead of action selection. Hence, our algorithm is more computationally efficient, as the number of steps $L_1$ in exploration is typically larger than the number of steps $L_2$ for the update of Q-functions. (e.g., in our experiments, we set $L_1 > 4L_2$).
> - As a result of the above reasons, our proposed OEB3 outperforms Chen et al. (2017) with BEBU evaluated by 49 Atari games.
> - In addition, our work establishes a theoretical connection between the proposed UCB-bonus $\rm{std}(Q)$ and the bonus-term $[\phi_t^\top\Lambda_t^{-1}\phi_t]^{\frac{1}{2}}$ in optimistic-LSVI that achieves a near-optimal worst-case regret, which motivates the design of OEB3.
>
> We revised the manuscript to add this discussion in Section 4.
>
> > Theorem 1 I think are well known in that the bonus used by UCB algorithms for linear regression is proportional to the posterior variance of bayesian linear regression. Further, LSVI-UCB is a frequentist solution approach and bootstrap DQN a non-parametric Bayesian approach — combining the two definitely warrants some discussion.
>
> (2) Despite the fact that the UCB-bonus is known to be consistent with the posterior standard deviation, utilizing such a bonus together with deep neural networks is scarcely explored. Although Chen et al. (2017) proposes to use a similar UCB-bonus in ensemble-Q learning, such utilization does not propagate uncertainty through time. Therefore, their approach does not capture the core benefit of utilizing UCB-bonus for the deep exploration of MDPs. We highlight that OEB3  propagates uncertainty through time by the backward update, which exploits the theoretical benefit of relevant analysis established by Jin et al. (2020). Moreover, we observe that such a connection between theoretical analysis and practical algorithm provides relatively strong empirical performance in Atari games and outperforms several strong baselines, which hopefully gives useful insights on combining theory and practice to the community.
>
> In addition, we highlight that Bootstrapped DQN is a non-parametric bootstrapped approach, which is a frequentist approach to construct the confidence interval of value functions. The output of such an approach coincides with the posterior under a Bayesian setting where the prior is uninformative (Friedman J. et al. 2001). In contrast, Optimistic LSVI constructs the confidence interval explicitly based on the linear model, which can be recovered by various bootstrap approaches such as the parametric bootstrap approach. Both Bootstrapped DQN and Optimistic LSVI construct valid confidence intervals for the value functions. In our approach, to fit in deep neural network parameterization, we adopt Bootstrapped DQN to calculate the standard deviation of Q-functions, which coincides with the bonus in Optimistic LSVI on the linear models.

---

### Author Response · Authors · 2020-11-23
**Summary of revision**

We thank the reviewers for their comprehensive and thoughtful comments. We carefully revised our submission based on the reviews, and hope the revisions address the reviewers' concerns.

In the updated manuscript we have made the following main changes:

- We added the detailed discussion of the difference between OEB3 and Chen et al (2017) in Related Works. (Section 4).

- We added the discussion of adjustments in OEB3 compared to optimistic-LSVI (Jin et al, 2020) and EBU (Lee et al, 2019). (Section 3.2).

- We added theoretical and empirical comparison of Uncertainty Bellman Equation and Exploration (O'Donoghue et al. 2018) and Bayesian-DQN (Azizzadenesheli et al. 2018). (Section 4 and Section 5.2).

- We added the related works of model-based approaches with optimism. (Section 4).

- We added additional algorithmic descriptions for three baselines (i.e., BEBU, BEBU-UCB, and BEBU-IDS). (Appendix B).

- We added empirical details of bootstrapping data in training and majority-vote in evaluation. (Section 5.1).

**References**

[Chen et al. 2017] Richard Y Chen, Szymon Sidor,  Pieter Abbeel,  and John Schulman. UCB Exploration via Q-Ensembles. arXiv preprint arXiv:1706.01502, 2017

[Jin et al. 2020] Chi Jin, Zhuoran Yang, Zhaoran Wang, and Michael I Jordan. Provably Efficient Reinforcement Learning with Linear Function Approximation. In Proceedings of Thirty Third Conference on Learning Theory. 2020

[Lee et al. 2019] Su Young Lee, Choi Sungik, and Sae-Young Chung. Sample-Efficient Deep Reinforcement Learning via Episodic Backward Update. Advances in Neural Information Processing Systems, 2019

[O’Donoghue et al. 2018] Brendan O’Donoghue, Ian Osband, Remi Munos and V. Mnih. The Uncertainty Bellman Equation and Exploration. International Conference of Machine Learning. 2018

[Azizzadenesheli et al. 2018] Kamyar Azizzadenesheli, Emma Brunskill, and Anima Anandkumar. Efficient Exploration through Bayesian Deep Q-Networks. Information Theory and Applications Workshop (ITA). 2018

---

### Decision · Program_Chairs · 2021-01-07
**Final Decision**

**Decision:**

Reject

**Comment:**

This paper represents a practical extension of sound theoretical uncertainty propagation ideas for exploration in deepRL. All the reviewers agreed this was a promising direction and the empirical results strong. It was nice to see additional qualitative analysis of the proposed method, beyond the typical "my number is bigger than yours" type of claims. The discussion was extensive; reviewers with specific subject matter expertise provided high quality and detailed reviews.  Several reviewers were in favour of the paper, but none were willing to champion it as a clear accept.

Indeed the discussion highlighted important concerns with the paper. Several reviewers found the paper to overclaim: most importantly the paper suggests strong theoretical underpinnings of the method without clear evidence. Some found the text very imprecise. The reviewers were torn if such changes represented wording changes or major rewrites. The original submission missed two key pieces of work which were added during the rebuttal phase---UBE was added by including the scores from the literature, and the other (Bayesian-DQN) was only added to the discussion. Good on the authors for doing so, though ideally both would be implemented again.

The AC's own reading of the paper highlighted a few other concerns. The writing needs work. In addition, the majority of improvement in overall performance appears to be due to very large improvements in a handful of games (e.g. Atlantis, Krull) and significant losses in performance in other games. This was not discussed at all. More surprisingly these games with huge performance gains were not used in the qualitative visualizations of the utility of the bonus found in the paper. The game breakout was used for analysis instead. Oddly the proposed method actually does worse or the same as SOTA methods (e.g., Adaptive EBU, Boot-DQN, UBE) in breakout.  This is difficult to get to the bottom of because: (a) the per-game score tables in the appendix don't include the scores achieved by important baselines (e.g., Boot-DQN, EBU, Adapt EBU), (b) different setups are used across the relevant literature (Boot-Q & UBE papers use 200m frames, EBU paper uses 10m frames, and this paper uses 20m), (c) the per-game analysis in the appendix focuses on comparing methods proposed in the paper under review. That might all make sense, but it is left to the reader to figure out and I never got to the bottom of it all (table 3 of the Lee et al contains some of the relevant comparison data). Such missing details and lack of analysis are particularly important when the paper boldly claims state of the art performance improvement.

All put together a clear picture emerges: the paper needs polishing, is unclear in places, over-claiming, and missing important analysis and explanations---in regards to both the theory and the experiments. The reviews are extensive and have provided many insights in how to improve the paper.